# Identifying antibiotics based on structural differences in the conserved allostery from mitochondrial heme-copper oxidases

Yuya Nishida [1,2], Sachiko Yanagisawa [3,14], Rikuri Morita [4,14], Hideki Shigematsu [5,13], Kyoko Shinzawa-Itoh[3], Hitomi Yuki [6], Satoshi Ogasawara[7], Ken Shimuta[8,9], Takashi Iwamoto[2], Chisa Nakabayashi[1,2], Waka Matsumura[3], Hisakazu Kato[2], Chai Gopalasingam[5], Takemasa Nagao[1], Tasneem Qaqorh [1,2], Yusuke Takahashi[1], Satoru Yamazaki[1], Katsumasa Kamiya[10], Ryuhei Harada [4], Nobuhiro Mizuno [11], Hideyuki Takahashi[8], Yukihiro Akeda[8], Makoto Ohnishi[8], Yoshikazu Ishii[12], Takashi Kumasaka [11], Takeshi Murata [7], Kazumasa Muramoto[3], Takehiko Tosha [5], Yoshitsugu Shiro [3], Teruki Honma[6], Yasuteru Shigeta [4], Minoru Kubo [3], Seiji Takashima[2] & Yasunori Shintani [1,2] ✉

Antimicrobial resistance (AMR) is a global health problem. Despite the enormous efforts made in the last decade, threats from some species, including drug-resistant *Neisseria gonorrhoeae*, continue to rise and would become untreatable. The development of antibiotics with a different mechanism of action is seriously required. Here, we identified an allosteric inhibitory site buried inside eukaryotic mitochondrial heme-copper oxidases (HCOs), the essential respiratory enzymes for life. The steric conformation around the binding pocket of HCOs is highly conserved among bacteria and eukaryotes, yet the latter has an extra helix. This structural difference in the conserved allostery enabled us to rationally identify bacterial HCO-specific inhibitors: an antibiotic compound against ceftriaxone-resistant *Neisseria gonorrhoeae*. Molecular dynamics combined with resonance Raman spectroscopy and stopped-flow spectroscopy revealed an allosteric obstruction in the substrate accessing channel as a mechanism of inhibition. Our approach opens fresh avenues in modulating protein functions and broadens our options to overcome AMR.

Antimicrobial resistance (AMR) is a global health problem[1]. Many efforts have been made to reduce the burden of AMR perils globally since 2013, yet threats from some species continue to rise regardless: drug-resistant *Neisseria gonorrhoeae* is one of five urgent threats[2,3]. Resistance to ceftriaxone, the last option for an empirical first-line antibiotic against *Neisseria gonorrhoeae* in most countries, has been reported and continues to emerge globally[4]. The gonococcal infection could become untreatable due to a high degree of AMR, which would increase serious complications: infertility, ectopic pregnancy, and increased transmission of HIV. The emergence of resistant pathogens to currently available antibiotics is very alarming; thus, the development of treatment options is imperative to tackle AMR.

The respiratory chain has recently garnered considerable scientific attention as a potential target for antibiotics. As a weapon to overcome AMR, compounds targeting the respiratory chain have been approved or entered clinical trials, for example, drugs against

---

parasites, fungi, and particularly drug-resistant *Mycobacterium tuberculosis*[5–10]. However, most of them are competitive inhibitors of orthosteric sites. As respiratory enzymes are essential for life, their core structure is generally conserved across species. Structural similarity and substrate commonality with host proteins are risks for cross-reactivity, which could be a cause of side effects[11]. Therefore, an allosteric inhibitor is a more feasible choice as allosteric sites are evolutionarily less conserved in amino acid sequence than orthosteric sites, theoretically improving selectivity and reducing toxicity[12,13]. However, a systematic and strategic search for allosteric inhibitors has not been established yet, especially against membrane proteins; most respiratory enzymes are membrane proteins.

HCOs are terminal respiratory enzymes present in all three domains of life: bacteria, archaea, and eukaryotes. HCOs receive electrons from the respiratory chain and reduce molecular oxygen to water. This exergonic reaction is coupled with proton pumping across the membrane, which contributes to maintaining the proton motive force that is further used for ATP production[14–18]. HCOs are multi-subunit complexes, and their constitution varies among species; however, subunit I is a catalytic subunit common in all HCOs. It contains a low-spin heme and a binuclear center (BNC), the catalytic site formed by a high-spin heme and a copper ion. The low-spin heme first receives electrons and transfers them to the BNC for the reduction of oxygen[16–19].

Eukaryotes originate from the symbiosis between Alphaproteobacteria and archaea[20]. Therefore, mitochondrial DNA-encoded subunits I to III of mitochondrial cytochrome c oxidase (mtCcO), analogous to eukaryotic HCOs, are descendants of the respiratory enzymes from bacteria[20,21], and functionally important residues, and thus their core structure is conserved, although the remaining residues are not the same. In addition, mammalian mtCcO has 10 more subunits encoded by genomic DNA. The physiological roles of these subunits are not fully elucidated[16]. The core structures of fundamental proteins such as RNA polymerases or ribosomes are also similar among species; intriguingly, they have acquired additional subunits that modulate their function along molecular evolution[22,23]. Thus, we hypothesized that the surface of the core structure of HCOs, which are covered by additional helices in mammals, might contain allosteric sites, regulating their activity positively or negatively. The existence of an additional helix in mammalian mtCcO makes the pockets distinct from bacterial HCOs.

In this work, we identify an allosteric inhibitory site buried inside eukaryotic mtCcO. The steric conformation around the binding pocket of HCOs is highly conserved among bacteria and eukaryotes, yet the latter has an extra helix. The structural difference in the conserved allostery enables us to rationally identify bacterial HCO-specific inhibitors: an antibiotic compound against ceftriaxone-resistant *Neisseria gonorrhoeae*.

## Results

### An allosteric inhibitory site buried inside mtCcO
To test this hypothesis, first, we need to identify an allosteric inhibitory site. We started with mammalian mtCcO inhibitors obtained by random compound screening. We have previously found that an endogenous protein directly interacts with mtCcO and allosterically modulates mtCcO activity[24,25]. This finding led us to perform random compound screening that modulates mtCcO activity; we identified mtCcO inhibitors, chemically distinct from the known inhibitors, including carbon monoxide, nitric oxide (NO), or cyanides. We selected several allosteric inhibitors after studying their enzyme kinetics (Supplementary Fig. 1a–d). We then attempted to obtain complex crystal structures of mtCcO and our inhibitors. We focused on T113 hereafter as its binding site was buried within a mammalian-specific helix, COX7C (Fig. 1a, b). X-ray diffraction data with a resolution of 2.2 Å were obtained from a mtCcO crystal soaked with T113. We also

determined the apo-structure of mtCcO under the same preparation conditions (Supplementary Table 1). The obtained complex structure showed a clear compound binding site that gave additional electron density compared with the protein, clearly found inside the mtCcO (Supplementary Fig. 2a). The $F_o$(T113)−$F_o$(DMSO) differential map confirmed that this electron density did not originate from water or lipid molecules, and showed the highest difference (Supplementary Fig. 2b). The binding pocket was different from the binding site for molecular oxygen or cytochrome c, the route for electron transfer pathway, proton pathway, or oxygen accessing channel[16,19], suggesting that T113 is a genuine allosteric inhibitor.

Three out of four helices surrounding the allosteric site of mtCcO belong to subunit I, common in HCOs. Furthermore, we noticed that the steric conformation of the helices near the low-spin heme is well conserved in bacterial HCOs (Fig. 2a and Supplementary Fig. 3). These observations led us to assume that we could reasonably screen allosteric inhibitors from derivatives of our mtCcO inhibitors, which likely target the corresponding allosteric site of bacterial oxidases.

### Rational identification of bacterial HCO-specific inhibitors
For the preparation of a custom library, we applied in silico compound screening originated from two mtCcO inhibitors, including T113 and T151, which we obtained in the initial high throughput screening; T151 was also found to bind the allosteric site (Supplementary Fig. 1d). Structurally similar compounds to those were primarily collected by multiple ligand-based search algorithms from 80 million commercially available compounds. Then, our in-house algorithm integrated them, ranked them, and chose the first series of 285 compounds[26,27]. We added the second set of 149 compounds chosen by docking simulation, which screened the same 80 million compounds against the allosteric pocket of mtCcO. In total, we established a custom library consisting of 434 compounds that have a high probability of binding to the conserved allosteric site for HCOs. We tested this library against mtCcO, and as a result, 47 compounds inhibited mtCcO more than 40% at 50 μM (11.4%), higher than the usual hit rate from random screening, verifying that our custom library concentrated mtCcO inhibitors (Supplementary Fig. 4a).

We used *E. coli* $bo_3$ ubiquinol oxidase ($bo_3$ UqO) to test our hypothesis as a model bacterial HCO. Our custom library was screened against $bo_3$ UqO, and we obtained 15 hit compounds that showed >40% inhibition for $bo_3$ UqO at 50 μM (Fig. 2b). Among them, eight common inhibitors for both mtCcO and $bo_3$ UqO, and more importantly, two specific inhibitors for $bo_3$ UqO, were successfully acquired (Fig. 2c). As expected, one of these inhibitors, N4, fitted well with the allosteric inhibitory curve assessed by the Michaelis–Menten equation (Fig. 2d).

To obtain direct evidence that N4 binds to the corresponding allosteric site, we determined the structure of N4-bound $bo_3$ UqO with a Fab fragment at 3.0 Å resolution using cryogenic electron microscopy (cryo-EM) (Fig. 3a, Supplementary Fig. 5e–h, and Supplementary Table 2). We also determined the apo-structure of $bo_3$ UqO at 3.1 Å under the same preparation conditions (Supplementary Fig. 5a–d). Differential maps between holo- and apo-structures demonstrated explicit additional density, and this density was the top difference found (Supplementary Fig. 5i). Notably, the binding site was exposed to the surface and adjacent to transmembrane helix 1 (TM1), TM2, and TM3 of subunit I, which strongly corroborates our hypothesis (Fig. 3b, c). There were hydrogen bonds between Asp75, Arg71, and N4. The electrostatic potential surface of the binding sites showed that $bo_3$ UqO has a more hydrophilic surface than mtCcO, suggesting it is unlikely that hydrophobic T113 binds to $bo_3$ UqO (Fig. 3d). Meanwhile, relatively hydrophilic N4 is not a feasible binder for the mtCcO allosteric pocket, a more hydrophobic environment, explaining that N4 is a derivative of T113, yet they are mutually exclusive inhibitors for $bo_3$ UqO or mtCcO, respectively. Mutants for

amino acid residues around the inhibitor in the structure demonstrated a significant change in the inhibitory effect, confirming that N4 definitively binds $bo_3$ UqO at the pocket (Supplementary Fig. 5j).

A specific inhibitor of bacterial HCOs might have the potential as an antibiotic. To test this possibility, we assessed the inhibition of *E. coli* growth by N4. In *E. coli*, there are two branches for terminal oxidases in the respiratory chain, $bo_3$ UqO and cytochrome *bd* oxidase (*bd* UqO). *E. coli* uses $bo_3$ UqO in aerobic conditions, whereas it preferentially uses *bd* UqO in hypoxic conditions, as an adaptation to the environment[28]. In wild-type *E. coli*, there was no effect of N4 in growth; however, it significantly decreased *E. coli* growth in $bo_3$ UqO-dependent strain (Fig. 3e). This growth inhibition was canceled in the $bo_3$ UqO knockout strain, confirming that the growth inhibition by N4 was a result of HCO inhibition, not a nonspecific effect. To distinguish between the bactericidal and bacteriostatic effects of the compound, we performed a colony count assay and found that the effect of our inhibitor, N4, on *E. coli* is bacteriostatic (Fig. 3f).

## Development of an antibiotic compound for *Neisseria gonorrhoeae*

Next, to expand our findings, we tested quinol-dependent NO reductase ($bb_3$ qNOR), a distant family member of HCO from the pathogenic bacteria, *Neisseria*. In the genus *Neisseria*, the emergence and spreading of multi-drug resistant *N. gonorrhoeae* is a considerable global health concern[29]. qNOR reduces NO to nitrous oxide ($N_2O$) and is a

critical enzyme in denitrification, which oxidizes nitrogen compounds to produce energy in anoxic conditions[30]. Bacterial denitrification also plays a vital role in protecting them from exogenous NO produced by host immune cells, thereby being implicated in the pathogenicity of several bacterial species, including *N. meningitidis* and *N. gonorrhoeae*[31,32]. These suggest that qNOR in *Neisseria* can be a drug target. We used *N. meningitidis* qNOR because the amino acid sequences of *N. meningitidis* qNOR and *N. gonorrhoeae* qNOR are 98% identical, especially 100% in the transmembrane region, and importantly the structure of *N. meningitidis* qNOR has been solved[33]. We confirmed that the steric conformation of three helices (TM 3–5 in $bb_3$ qNOR) around the low-spin heme is also conserved in $bb_3$ qNOR (Fig. 4a)[34]. We employed the same approach to $bb_3$ qNOR as for $bo_3$ UqO. First, we screened our custom library against $bb_3$ qNOR and obtained 52 hit compounds that showed >50% inhibition of $bb_3$ qNOR at 50 μM (Fig. 4b). Among them, we found one $bb_3$ qNOR-specific inhibitor, Q275. Q275 did not cross with either mtCcO or $bo_3$ UqO (Fig. 4c). Q275 did not affect the cellular respiration of mammalian cells, nor cell viability of them (Fig. 4d, e). We confirmed that Q275 has an allosteric mode of inhibition (Fig. 4f). Further, amino acid-substituted mutants around the corresponding allosteric site of $bb_3$ qNOR showed a considerable change in the inhibitory effect, confirming that Q275 binds to $bb_3$ qNOR at the pocket (Fig. 4g).

Finally, we tested the antimicrobial effect of Q275 against a clinically isolated strain of *N. gonorrhoeae* that has resistance to ceftriaxone

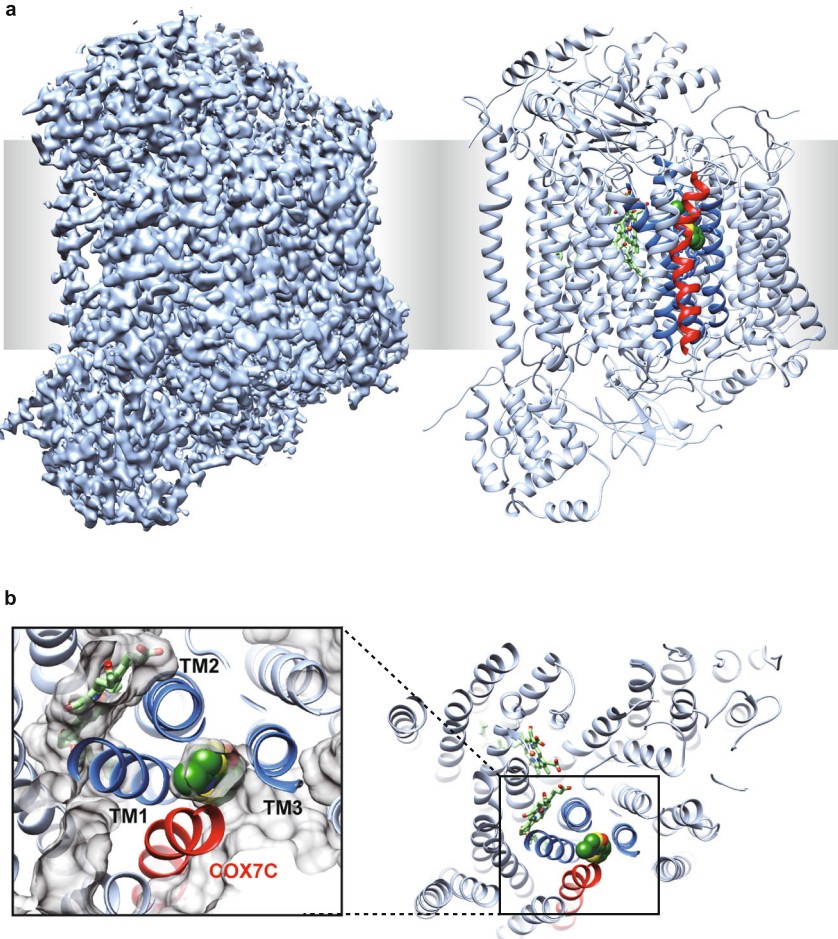

**Fig. 1 | The allosteric site for T113 is buried inside eukaryotic mtCcO. a** X-ray structure of mtCcO with T113. The electron density map ($2F_o−F_c$), contoured at 1σ, is shown in the left. The ribbon model of mtCcO with T113 in the sphere is shown on the right. T113 was covered by COX7C (red), hidden from the surface. **b** T113 was surrounded by 4 transmembrane helices (TM1–3, COX7C) and buried from the surface, viewed from the inter-membrane space. Protein molecular surface is shown as gray in the close-up view. Three helices of subunit I surrounding the allosteric site are shown as dark blue, the other helices of subunit I as pale blue, subunit COX7C in mtCcO is shown as red.

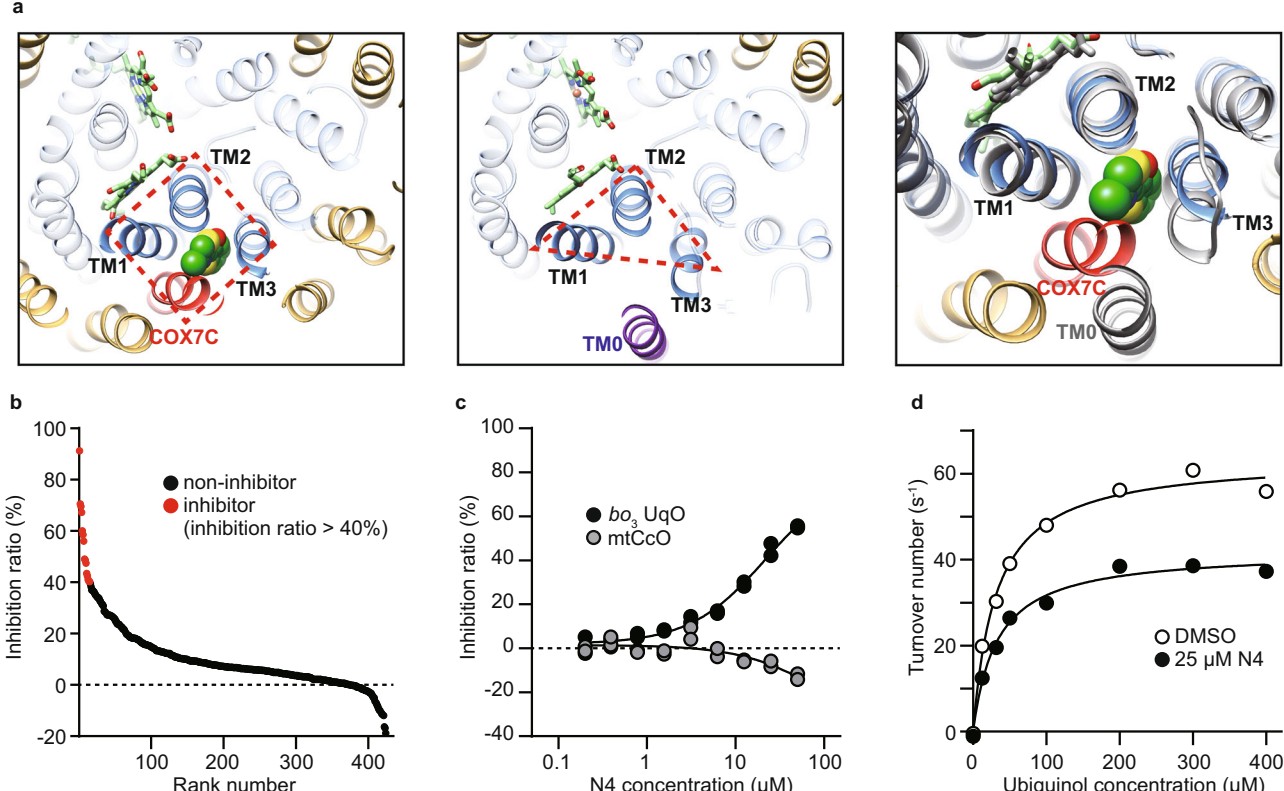

**Fig. 2 | Identification of bacterial HCO-specific allosteric inhibitors. a** Allosteric sites of mitochondrial mtCcO (left) and *E. coli bo₃* UqO (center). Helices of subunit I are shown as blue, TM0 of subunit I as purple, subunit COX7C is shown as red, and the other helices as yellow. Merged views are shown (right) with *E. coli bo₃* UqO in gray. COX7C is close enough to cover the inhibitor. All the helices in subunit I are merged between mtCcO and *E. coli bo₃* UqO except TM0. **b** Screening of 434 chemicals at 50 μM against *E. coli bo₃* UqO. Data are presented as an average of a duplicate. **c** Dose-dependent inhibition of N4 specific on *bo₃* UqO enzymatic activity. Data are presented as an average value of technical replicates over two independent experiments. **d** Kinetic analysis of *bo₃* UqO with DMSO and 25 μM N4 molecule. Fitting lines are calculated by using Michaelis–Menten equation with the non-competitive inhibition model. Data are presented as an average value of technical replicate. Reproducibility was confirmed by two independent experiments. Source data are available as a Source Data file.

(FC428); the spread of the super-resistant *N. gonorrhoeae* has become a serious global health problem[4]. Notably, Q275 demonstrated antimicrobial effects against FC428 as well as the reference strain (WHO F) in NO-challenging (attack from immune cells) conditions, mimicking the infectious intravital environment (Fig. 4h). We established *norB*-deficient, encoding qNOR, strains in both WHO F and FC428 backgrounds. In NO-challenging condition (20 mM NaNO₂), *norB*-deficient strains in both WT and FC428 did not grow, further supporting that growth inhibition is mediated via qNOR. To distinguish between bactericidal and bacteriostatic effects using these *norB*-deficient strains, we performed colony count after exposure to NaNO₂ and found that targeting qNOR in *N. gonorrhoeae* is bacteriostatic (Fig. 4i). These results suggest that our approach developed a specific inhibitor against a pathogenic bacterial HCO on-demand with a narrow range of specificity, having the therapeutic potential to tackle AMR.

**Allosteric inhibition in the substrate accessing channel**
We next focused on the inhibitory mechanism. The binding site of T113 on mtCcO formed a narrow tunnel surrounded by four helices in total (Fig. 1b). The $F_o$(T113)−$F_o$(DMSO) differential map revealed several differences in the structure of mtCcO triggered by the binding of the inhibitor: the exit site of the proton pathway (Asp50, Asp51), a side chain of Ser382 of TM10 in subunit I, and hydroxyfarnesylethyl group of heme *a* (Supplementary Fig. 2b). These are the sites where structural change has been reported between the reduced and oxidized forms of mtCcO, raising the possibility that T113 might have reduced mtCcO[35]. To test this possibility, we analyzed the signature of mtCcO reduced through dithionite or by T113. Compared to a fully reduced state

induced by dithionite, mtCcO mixed with T113 showed a minimum signature of reductive change in the Soret band and an increase in around 600 nm absorption spectra, which are hallmarks of reduced hemes, suggesting that T113 is not a reductive reagent (Supplementary Fig. 6). Therefore, the structural change we observed does not fully explain the mechanism of inhibition.

In the mechanism of allostery, an effector binding transmits the signal to the functional site, the orthosteric site, by a transition of conformational ensembles, which is often difficult to capture by structural analysis because of its snapshot nature or possible constraint in crystallization[36]. Therefore, to obtain a mechanistic insight, we applied molecular dynamics (MD) simulation of mtCcO with or without the inhibitor. Overall, MD simulations with the inhibitor (holo-MD) did not show significant structural deformation. Notably, the average trajectories with the inhibitor showed that TM2 of subunit I, one of the four helices forming the inhibitor binding pocket, bent to the direction of TM5/6, although TM3 did not change between the holo- and apo-MDs (Fig. 5a–c). This movement suggests that the channel for molecular oxygen, the hydrophobic cavity surrounded between TM2/4 and TM5/6, is constricted. The holo-MD demonstrated that the distance between Glu242 and Ile66, both of which face the oxygen channel and form a minimum cross-section part, became narrower in the presence of the inhibitor. In contrast, the distance between Ile312 and Val287, which face an exit path for the produced water, did not change (Fig. 5c, d). Furthermore, we performed additional simulations with ligand removal after ligand-bound MD (Re-apo MD). Re-apo MD demonstrated that

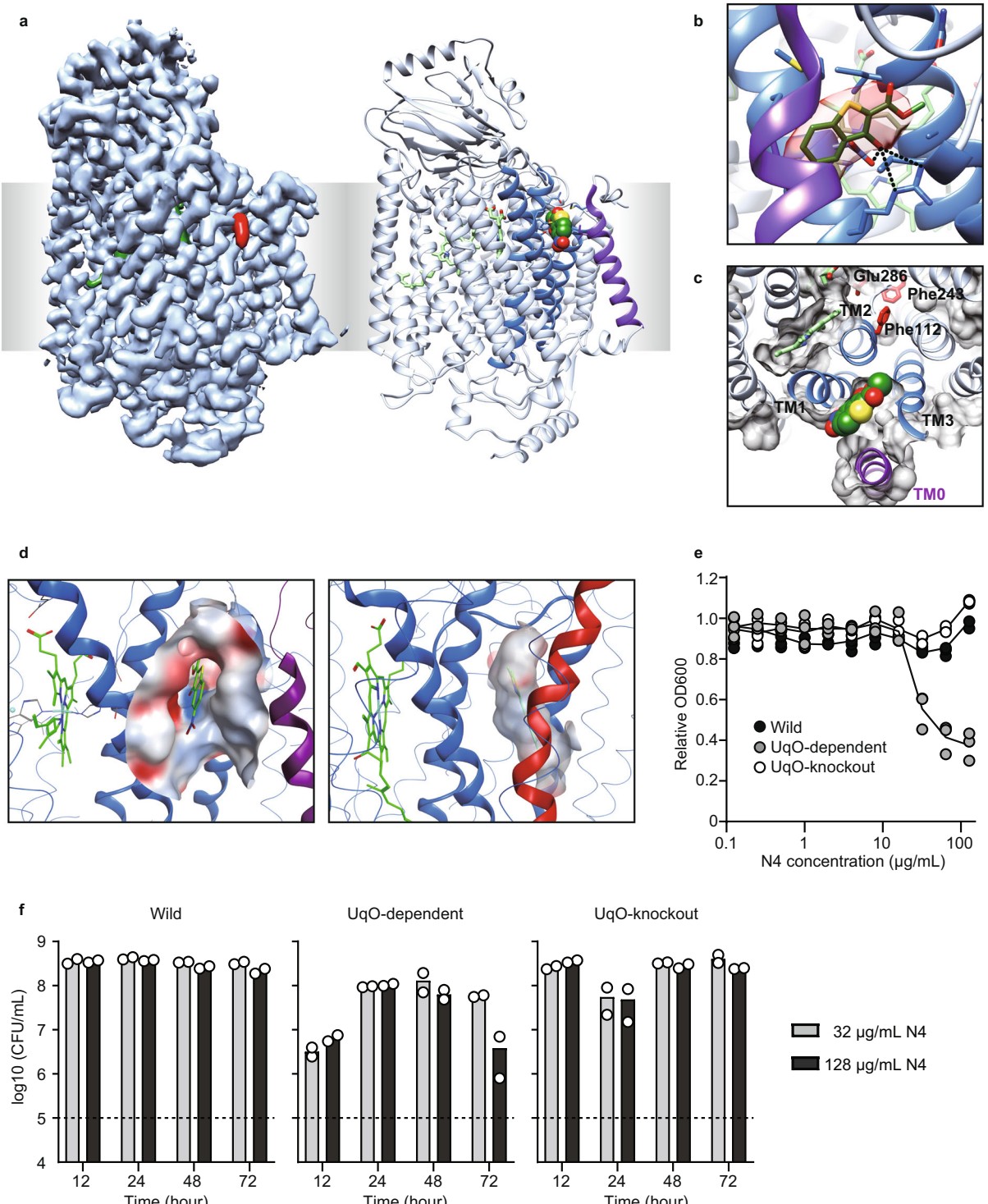

**Fig. 3 | Cryo-EM proof of the allosteric site of *bo*₃ UqO. a** Cryo-EM density map (left) and the ribbon model (right) of *bo*₃ UqO in complex with N4 (red in the left, sphere in the right). The allosteric site of *bo*₃ UqO is exposed. **b** N4 molecule in the allosteric site of *bo*₃ UqO. The cryo-EM density map around N4 is shown in red. Molecular interactions between Asp75, Arg71, and N4 are shown as dotted lines. **c** N4 is accessible from the surface. The protein molecular surface at the plane of Fe in heme *b* is shown as gray. The whole N4 molecule was presented. **d** The electrostatic potential surface of the allosteric site of *bo*₃ UqO (left) and mtCcO (right). **e** Growth inhibition by N4 for 24 h culture of wild-type *E. coli* (Wild), *bo*₃ UqO-knockout strain (Δ*bo*₃), and *bo*₃ UqO-dependent strain (Δ*bd*). **f** A time- and concentration-dependent viability assay. A dotted line indicates inoculation (1 × 10⁵ CFU/ml). Data are confirmed by biological triplicate (**e**) or duplicate (**f**). The color of the helices are as same as in Fig. 2. Source data are available as a Source Data file.

TM2 relaxed to the initial apo structure, although the change in the oxygen channel did not relax during the Re-apo MD, suggesting that the effect on TM2 of the inhibitor that is next to TM2 is more direct than on the oxygen channel. These data suggested

that T113 might interfere with oxygen access to the BNC, thereby inhibiting enzyme reaction.

To test whether T113 has an allosteric effect in the oxygen channel, we performed resonance Raman spectroscopy, a sensitive

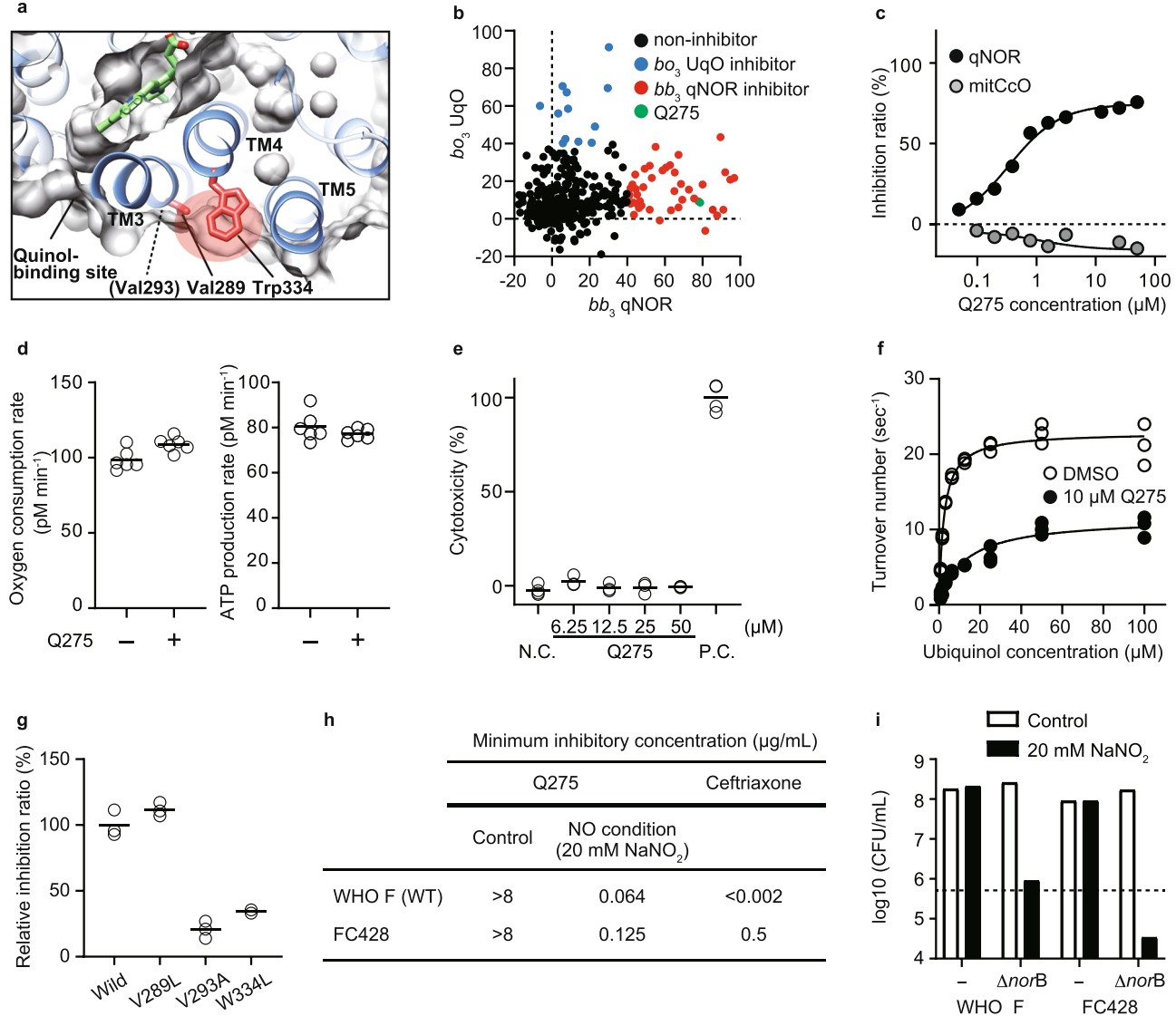

**Fig. 4 | Identification of *N. gonorrhoeae bb*₃ qNOR-specific inhibitor. a** The steric conformation around the low-spin heme is conserved in $bb_3$ qNOR (PDB: 6fwf). A red shadow indicates the allosteric site, distinct from the quinol binding site. **b** Screening of 434 chemicals at 50 μM on *N. meningitidis* $bb_3$ qNOR (*x*-axis) and *E. coli* $bo_3$ UqO (*y*-axis). Q275 is a specific $bb_3$ qNOR inhibitor, shown in green. **c** Dose-dependent and specific inhibition of Q275 on $bb_3$ qNOR activity. **d** Q275 has no effect on oxygen consumption rate in mammalian cells. *N* = 6 technical replicates for each group and reproduced in two independent experiments. **e** Q275 shows no cytotoxicity assessed in rat cardiomyocytes. N.C.; DMSO, P.C.; 1% Triton X-100. *N* = 6 technical replicates for DMSO, 3 for other groups. Reproducibility was confirmed by two independent experiments. **f** Kinetic analysis of $bb_3$ qNOR with DMSO

and 10 μM Q275 molecule. Fitting lines are calculated by using Michaelis–Menten equation with the non-competitive inhibition model. **g** The effect of amino acid substitution of $bb_3$ qNOR. The positions of Val293, Val289, and Trp334 are shown in (**a**). **h** Minimum inhibitory concentration (MIC) of Q275 against the *N. gonorrhoeae* reference strain WHO F and a ceftriaxone-resistant FC428. Sodium nitrite (20 mM) was added to mimic a NO condition. The data represent duplicate experiments. **i** Colony count 24 h after Q275 treatment of the *N. gonorrhoeae* WHO F, FC428, and *norB*-deficient from both strains. Sodium nitrite (20 mM) was added to mimic a NO condition. A dotted line indicates inoculation (5×10⁵ CFU/ml). **c**, **f**, **g** Data are presented as an average value of technical replicates over three (**f**, **g**) independent experiments. Source data are available as a Source Data file.

method for detecting structural changes that cannot be assessed by X-ray crystal structural analysis. As T113 has autofluorescence, we screened its derivatives and chose N62 because of its higher affinity without autofluorescence (Supplementary Fig. 4b, c). We confirmed that N62 bound the same binding site in co-crystallography (Supplementary Fig. 4d) and gave a small difference around 440 nm in the heme absorption similar to that with T113 (Supplementary Fig. 4e). We used N62 for Raman spectroscopic analyses hereafter. Previously, it was reported that visible light induces photoreduction of mtCcO, where heme *a* is initially reduced, followed by heme $a_3$[37]. Figure 4e depicts the resonance Raman spectra of mtCcO with and without N62, focusing on the photoreduction of the hemes. The resonance Raman band at 1356/1372 cm⁻¹ is assignable to the $v_4$ mode of the

hemes (heme *a* and heme $a_3$), an indicator of the redox status: 1356 cm⁻¹ for the reduced state, 1372 cm⁻¹ for the oxidized state[38]. In control conditions without N62, laser irradiation at a laser power of 1 mW demonstrated photoreduction of hemes. When the irradiation was stopped, available oxygen caused restoration of the oxidized hemes, reversing the redox marker bands. Reirradiation with the laser demonstrated a reduction of the hemes again (Fig. 5e(i)). By contrast, mtCcO with N62 displayed a different photoreduction behavior. With N62, a comparable reduction of the hemes was observed with a low laser power of only 0.1 mW. Notably, discontinuation of laser irradiation did not decrease the reduction marker band and reirradiation exhibited a further increase of it (Fig. 5e(ii)). These data indicate that N62 inhibits reoxidation of the

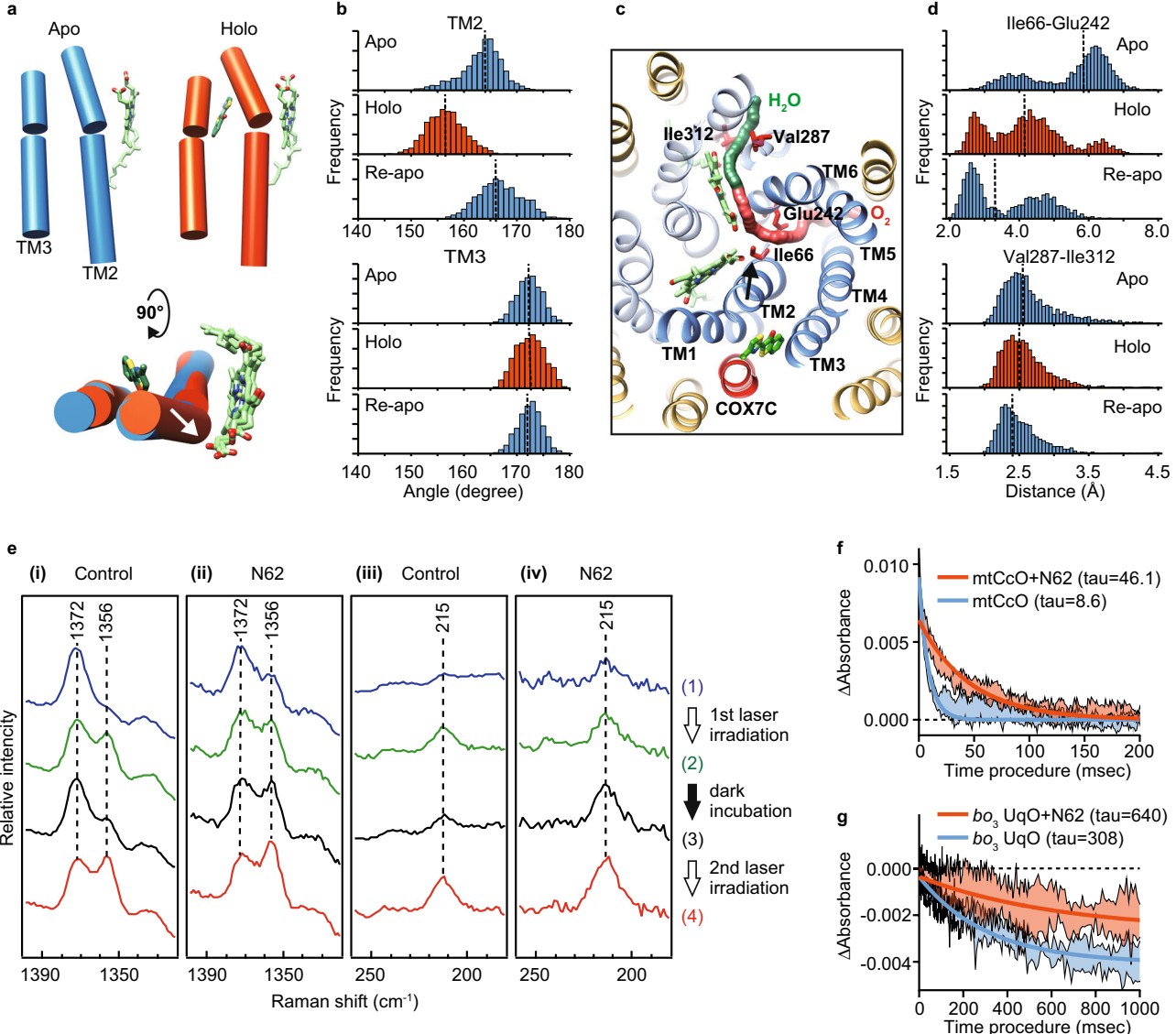

**Fig. 5 | Allosteric obstruction of oxygen access to the BNC as a common mechanism of inhibition. a** Axes of TM2 and 3 in a representative snapshot from a trajectory of apo-MD, holo-MD, and Re-apo MD. The axes were calculated by UCSF Chimera tools and represented by cylinders. Heme $a$ and T113 molecules were shown as green sticks. **b** Distribution histograms of the angle for TM2 and 3. **c** The focused region in the MD simulation. Paths of the oxygen channel (red) and the water channel (green) were calculated by CAVER and shown as continued spheres. TM1-6 in subunit I is shown as dark blue, and the other helices of subunit I as pale blue, subunit COX7C in mtCcO is shown as red, and the other helices as yellow. **d** Distribution histograms of the distance between Glu242-Ile66 in the oxygen channel and the distance between Ile312-Val287 in the water channel. Arrows show the movement of TM2 by T113 in (**a**, **c**). Dot lines in **b** and **d** show the median value. **e** The 441.6-nm excited resonance Raman spectra of mtCcO hemes ((i), (iii)) without

photoreduced hemes by oxygen. To probe the oxygen binding more directly, we next analyzed the Fe-His stretching mode ($\nu_{\text{Fe-His}}$) of heme $a_3$, the oxygen binding site, at 215 cm$^{-1}$, representing the 5-coordinated, reduced state of heme $a_3$. The 215 cm$^{-1}$ bands will disappear when molecular oxygen binds to heme $a_3$. mtCcO without N62 exhibited an increase/decrease cycle of the 215 cm$^{-1}$ band (Fig. 5e(iii)); however, mtCcO with N62 demonstrated a continuous rise in the 215 cm$^{-1}$ band (Fig. 5e(iv)), which provides an experimental evidence that N62 inhibits the binding of oxygen to heme $a_3$. Inhibition of reoxidation of the heme $a_3$ by oxygen lowers the threshold of the photoreduction, indicating that the structural change in

and ((ii), (iv)) with N62. mtCcO sample was laser irradiated twice with 10 min dark incubation in between. Spectra ((1); blue) and ((2); green) were obtained from 0 to 3 and 27–30 min in the first 30-min irradiation. After 10 min dark incubation, spectra ((3) black) and ((4) red) were obtained from 0 to 3 min and 27–30 min in the 2nd 30-min irradiation. The laser power was 1 mW for ((i), (iii)) and 0.1 mW for ((ii), (iv)). **f** A stopped-flow experiment showed that N62 inhibited the CO binding to CcO five times slower than the control. Absorbance changes at 440 nm are associated with the binding of CO to the reduced mtCcO. **g** Absorbance changes at 430 nm associated with binding of CO to the reduced wild-type $bo_3$ UqO. Fitting lines are calculated by a one-phase decay model. The range of data is presented as a standard deviation of four replicates for CcO, and three replicates for $bo_3$ UqO. Source data are available as a Source Data file.

mtCcO crystal was presumably caused by X-ray-induced reduction during the data acquisition.

To further strengthen the inhibitor's effect on the oxygen channel, we performed a stopped-flow experiment in which we could directly assess the access of carbon monoxide (alternative to molecular oxygen) to the binuclear center[39]. As shown in Fig. 5F, N62 inhibited the CO binding to CcO five times slower than the control. Therefore, we concluded that allosteric inhibition in the oxygen-accessing channel plays a major role in their inhibition of mtCcO.

It is plausible that the allostery we found in mtCcO is also preserved between bacterial HCOs and their inhibitors. To this end, we created a

single amino acid substitution in the amino acids that face the oxygen channel of $bo_3$ UqO, which can be genetically manipulated. Among the mutants made, less bulky substitutions of Glu286 and Phe112, which correspond to Glu242 and Phe67 in mtCcO (Figs. 3c and 5c), reduced the inhibitory effect of N4 (Supplementary Fig. 6c). Further to obtain direct evidence, we performed a stopped-flow experiment with $bo_3$ UqO. As N4 has absorbance around 400–450 nm, we used N62 for this experiment. N62 is a derivative of T113 and a common inhibitor for both mtCcO and $bo_3$ UqO. First, we confirmed that the inhibitory effect of N62 on $bo_3$ UqO was also reduced by the mutants in the oxygen channel as same as N4 (Supplementary Fig. 6d). A stopped-flow experiment with $bo_3$ UqO demonstrated that the inhibitor also slowed the CO binding in $bo_3$ UqO, as it did on mtCcO (Fig. 5g). Collectively with these multimodal analyses, we conclude that our HCO inhibitors allosterically obstruct molecular oxygen/NO entry to the BNC by a conformational change of a transmembrane helix of subunit I, thereby inhibiting HCO function.

## Discussion

Drugs with a unique mode of action are critical for expanding our antimicrobial options to overcome AMR, especially in cases where the pathogens, including *N. gonorrhoeae*, have acquired resistance to all the currently available antibiotics, spreading globally, and would become untreatable[40]. The respiratory chain has been a prospective target for antibiotics development; allosteric inhibitors are more desirable than orthostatic ones to minimize the risk of side effects, considering the importance of the respiratory chain in life. Here, we identified conserved allostery in HCOs and an additional helix that is only found in eukaryotic mtCcO. This structural difference in the allostery allowed us to isolate the specific inhibitors for two different bacterial HCOs, including an antibiotic against a ceftriaxone-resistant *N. gonorrhoeae*, which is one of five urgent threats in the 2019 report from the Centers for Disease Control and Prevention[3].

This study is a proof of concept, and the compound is still in the early stage of drug development; however, our findings will pave the way for the development of antibiotics with a different mechanism of action. The comparison of the published structures of HCOs revealed that the steric conformation around the low-spin heme is very much conserved in all HCOs from bacteria, yeasts, plants, and mammals (Supplementary Fig. 3)[41], suggesting that the allosteric site identified by us is likely to be conserved. Hence, our approach can generate specific inhibitors, potential antibiotics, for each bacterial HCO on-demand with a narrow range of specificity. The development of narrow-spectrum agents is in line with the current requirement for minimizing the effect on the host microbiome and preventing widespread resistance[42]. Targeting HCOs may not be simple, as pathogenic bacteria often have multiple terminal oxidases in their respiratory chain. Therefore, a desirable HCO inhibitor as an antibiotic is the one that specifically targets a particular infectious stage where pathogens critically require the HCO to adapt to the growing environment[43], as we have shown qNOR as a therapeutic target for *N. gonorrhoeae* in this study, or can be used with other antibiotics as a combination therapy. Our results suggest that the effects of UqO inhibitor on *E. coli* and qNOR inhibitor on *N. gonorrhoeae* were both bacteriostatic. Although bactericidal action sounds preferable, the superiority of bactericidal action over bacteriostatic has rarely been documented[44]. Further research and development of HCO inhibitors are necessary to reach the clinical arena.

One of the four helices in mtCcO surrounding the allosteric pocket is the genome-encoded subunit COX7C, covering the surface as if it hid the allosteric site. Our phylogenic analysis indicated that the allosteric site of the ancient HCOs was exposed, and it has been sealed in eukaryotic, mitochondrial HCOs during molecular evolution (Supplementary Fig. 7). The eukaryotic-specific subunit might protect mtCcO from access to inhibitors, or there might be an endogenous inhibitor that negatively regulates mtCcO activity at a specific time. The evolutionary role of the acquired subunit requires further research.

## Mechanisms of inhibition

Combined with stopped-flow experiments, resonance Raman spectroscopy, and mutant analysis, we concluded that our HCO inhibitors allosterically obstruct molecular oxygen/NO entry to the BNC by a conformational change of a transmembrane helix of subunit I, thereby inhibiting HCO function. However, notably in the case of $bo_3$ UqO, the site N4 binds is the quinol binding site. The quinone-$bo_3$ UqO structure confirms that N4 occupies the space where the substrate binds[45]. Asp75 and Arg71 are the same amino acids N4 used for molecular interaction as quinone does[45], suggesting that N4 inhibits $bo_3$ UqO by obstructing the substrate binding. These findings suggest that the space surrounded by TM0 and TM1–3 of $bo_3$ UqO works as both a substrate binding site and an allosteric inhibition site we proposed. TM0 is only present in quinol oxidases, including $bo_3$ UqO. It effectively stabilizes hydrophobic ubiquinol in the transmembrane region so that $bo_3$ UqO can use as a substrate; the existence of TM0 makes the ubiquinol oxidase family unique. TM0 is not found in other types of HCOs, in which the allosteric site we proposed is distinct from the substrate binding site, as shown in qNOR which does not have TM0.

Regarding mtCcO, we demonstrated that the inhibitory mechanism on oxygen entry plays a significant role in T113 and N62 inhibition for mtCcO; however other inhibitory mechanisms might also be involved, as in the case we discussed for N4 on $bo_3$ UqO. We found the structural change in Asp50/51 and Ser382 in the mtCcO-T113 crystal structure. We reasoned that it was caused by photoreduction during sample preparation and radiation-induced reduction for the following reasons. (1) change in Asp50/51 and Ser382 is found in the reduced form of CcO structure; however, the simple addition of T113 did not cause the reduction of CcO (Supplementary Fig. 6a, b). (2) laser irradiation during Raman data acquisition caused CcO reduction. Figure 5e suggests that the mtCcO inhibitor lowered the threshold of photoreduction. (3) MD simulation with the inhibitor did not cause the structural change in Asp50/51 (Supplementary Fig. 6e). These observations, however, did not eliminate the possibility that the binding of T113 induces the structural change found in Asp50/51 and Ser382. The change in these residues might affect proton pumping or electron transfer; Asp50/51 and Ser382 are essential residues for proton pumping, especially in mtCcO, forming the H channel as suggested by Yoshikawa et al.[16], although H channel is only found in mtCcO. Rich and colleagues indicated that H channel works as dielectric well[17]. Furthermore, the Sharma group recently reported that conformational change in the Ser382 carrying domain affects electron transfer[46]. Thus, perturbation in the region may cause inhibition of proton pumping or electron transfer. Further study is warranted.

Our MD simulation of 100 ns gave us an important clue to the allosteric mechanism of the mtCcO inhibitor; however, Re-apo MD did not show that the oxygen channel relaxes apo. Longer MD may be needed to clarify molecular mechanistic details.

## Future perspective for finding allosteric modulators

Our approach can be applied to finding allosteric modulators in other therapeutic targets. Enzymes generally acquire additional domains or subunits along molecular evolution[22,23]. As the respiratory chain in bioenergetics is fundamental and essential for life, respiratory enzymes other than HCOs are also conserved among species, and their core structures too. The size of the molecule of these respiratory enzymes is bigger in eukaryotes than in their bacterial counterparts. They could likely contain allostery inside the protein at the boundary of the structures between eukaryotes and bacteria, leading to the development of antibiotics, as the respiratory chain is a proven target for antibiotics. Furthermore, any fundamental molecule essential for life and conserved among species could be a potential target. Also, the

additional peptides might contain a positive allosteric site at the border of their core structure; a positive allosteric modulator for the loss-of-function human disease could be a therapeutic direction.

Generally, the search for an allosteric site is challenging, requiring considerable experimental work for each target protein and difficult to apply to others. The concept of buried conserved allostery will help develop a systematic approach, which has been critically desired. The number of protein structures has been considerably increased by the emergence of cryo-EM and recent advances in structural prediction such as Alphafold2 and RoseTTAfold[47,48]. Comparing protein structures among species, not only static structures but also ensembles generated by MD will accelerate finding buried conserved allosteric sites. Thus, in conclusion, this study will open fresh avenues in protein science and therapeutic development, especially for antibiotics with different mechanisms of action.

## Methods

### Preparation of resting oxidized bovine heart mtCcO crystals
Bovine heart cytochrome c oxidase solution and crystals were prepared as described in a previous study[49]. Before freezing, the crystals were treated with 2 mM compound or DMSO in the final medium.

### X-ray diffraction experiment and structural determination
X-ray experiments were carried out at SPring-8 beamlines BL26B1/B2. The diffraction data processing and scaling were carried out using XDS[50]. The initial phase was calculated by MOLREP[51] using a model of PDB code 5B1A after removing nonprotein molecules. To improve the density and remove the model bias, the maximum entropy method in Phenix[52] was performed. For the calculation of differential electron density map between with and without compound, $F_o$–$F_o$ map calculation in CNS[53] was used with a ligand-omitted model. Rebuilding was performed by using COOT[54]. Models were refined with REFMAC5[55] and phenix.refine[56] with the atomic parameters revised by Dr. Tomitake Tsukihara. To remove the model bias, all electron density maps used in rebuilding were calculated with the compound-omitted model. The electron density maps were confirmed with replicate samples. Refinement statics are provided in Table S1.

### Enzyme assay of mtCcO
Purified mtCcO and compounds were incubated on 96-well plates at 30 °C for 30 min in assay buffer (pH 7.4 50 mM potassium phosphate, 0.1% bovine serum albumin, 0.025% 14:0 Lyso PG). To start the enzymatic reaction, reduced cytochrome c (1.5 μM) was added to the mixture, then the plates were read at 550 nm using a plate reader. The slope for 60 s was calculated and the change in mtCcO activity was exported by comparing it to the DMSO-treated sample as a negative control.

### Absorption and resonance Raman spectroscopy
mtCcO solutions for spectroscopy were prepared to 15 μM mtCcO with and without 50 μM N62 in 60 mM sodium phosphate buffer and 0.2% n-decyl-β-ᴅ-maltoside (pH 6.8) and their spectra were measured at 4 °C unless otherwise noted. Absorption spectra were measured in a 3 mm path cuvette with a spectrophotometer (lamda650, PerkinElmer). The spectra were presented with 1-cm equivalent absorbance values. Oxidized mtCcO was measured under air and dithionite-reduced mtCcO was measured under $N_2$. Resonance Raman spectra were measured in a spinning cell at 2000 rpm with the excitation wavelength at 441.6 nm (a HeCd laser, Kimmons). Raman scattered light was detected by a Raman spectrometer (500M, SPEX) attached with CCD (Symphony II, Horiba) with a 90° scattering geometry. Resting oxidized mtCcO solution for Raman measurements was prepared and transferred into the spinning cell sealed with a rubber septum in a glove box (MM3-H60S, MIWA) at an oxygen level below 100 ppm. The used buffer and rubber septum were degassed and incubated in the glove box overnight. The sample solution thus prepared contained only a small amount of residual oxygen, which ensured the observation of a five-coordinate, reduced heme $a_3$ fraction in the photoreduction measurement, while avoiding immediate full oxidation of all heme $a_3$.

### Molecular dynamics simulations
The model system was constructed from the reduced bovine mtCcO crystal structure (PDB ID code 3AG2) to perform all-atom MD simulations. All crystal water molecules present in the PDB ID code 3AG2 atomic structure were removed. The missing hydrogen atoms of the systems were added with the tleap program in AmberTools 2020 (ref: https://ambermd.org/CiteAmber.php). The Amber ff14SB force field for protein, water, and ions[57], and Amber lipid14 force field for lipids[58] were adopted in the MD simulations. The entire 13-subunit monomeric mtCcO was immersed in a lipid bilayer formed by 50% phosphatidylcholine and 50% phosphatidylethanolamine using CHARMM-GUI[59]. The protein with lipid bilayers was then solvated with the TIP3P water model in rectangular boxes. The boxes were prepared with margins of 20 Å apart from each protein to avoid artificial effects by its periodically repeated images. The parameters for metal centers and amino acids were obtained from an earlier study[60]. Briefly, $Cu_A$ and heme a were oxidized and heme $a_3$ and $Cu_B$ were reduced. The protonation states of residues were determined by the electrostatic continuum method. The protonation states of the important residues are as follows: all the propiones of hemes a and $a_3$ were deprotonated, Arg438 and 439 were de-protonated, Asp364 and Glu242 were de-protonated, His290 and His291 were protonated in the δ-position. To neutralize each system with counterions, some water molecules were randomly selected and replaced with a set of $Na^+$ or $Cl^-$ ions. The particle-mesh Ewald (PME) method was employed for Coulomb interactions[61]. All the MD simulations were performed using the GROMACS 2019 software[62]. The entire membrane–protein–solvent system consisted of 302,004 atoms for apo-MD, and 302,029 atoms for holo-MD. The MD simulations were performed under constant-$NPT$ (300 K and 1 atm) ensemble using 2 fs time step.

The following treatments (a–c) were considered to equilibrate each system: (a) Energy minimization was performed for 10,000 steps. (b) A 100 ps MD equilibration was performed with the V-rescale thermostat[63] at 300 K under 1 atm, and (c) a 100 ps MD equilibration was performed with the Berendsen coupling[64] at 300 K under 1 atm. Each snapshot was recorded every 1 ps[65]. Finally, a 100 ns MD simulation was performed under constant-$NPT$ (300 K and 1 atm) ensemble as production runs and the last 80 ns were used as analyses. To obtain statistically reliable trajectories, we performed three runs with different initial velocities for the apo or holo-structures.

### Preparation of a custom compound library
For the first group, we calculated the Tanimoto coefficient between four mtCcO inhibitors and each of the 80 million commercially available compounds on the basis of 2D fingerprints (MACCS keys, ECFP4, FCFP4, and GpiDAPH3) and 3D shape metrics (ComboScore) with the software Pipeline Pilot (BIOVIA, Dassault Systèmes), MOE (Chemical Computing Group), and ROCS (OpenEye Scientific Software). Compounds showing high Tanimoto coefficient values were clustered by ECFP4 fingerprint, and then compounds in the center of each cluster were selected for enzyme assay. For the second group, mtCcO/T113 complex structure was prepared appropriately using a preparation wizard in Schrödinger suite 2016-1 (Schrödinger, LLC), and then a receptor grid generation was performed for the docking protocol. 80 million commercially available compounds were subjected to molecular docking against mtCcO using Glide 7.0. Compounds selected based on the Glide score, which shares the same space as T113 in the complex, were clustered by ECFP4 fingerprint, and then compounds in the center of each cluster were selected for enzyme assay. Finally, 434 compounds were selected and purchased.

## Preparation of $bo_3$ UqO

The plasmid coding $bo_3$ UqO with a carboxyl-terminus His-tag on subunit II was a gift from Gennis laboratory[66]. $bo_3$ UqO was purified based on the method as described previously[67]. Briefly, the plasmid was transformed into the C43 (DE3) $\Delta cyo$ E. coli strain, and cells were grown in an M63 medium. $bo_3$ UqO was solubilized by 1% n-dodecyl-β-D-maltoside (C12M) and purified with TALON resin, Super-Q resin, and size-exclusion chromatography, and then stored in a buffer containing 25 mM Tris–HCl, 200 mM NaCl, and 0.02% C12M (pH 7.5) at a concentration of 10–15 mg/mL.

## Establishment, screening, and preparation of monoclonal antibody

The purified E. coli $bo_3$ UqO was reconstituted into liposomes (egg yolk phosphatidylcholine: E. coli polar lipid = 3:1). Female BALB/c mice were immunized five times with 0.1 mg doses of the reconstituted $bo_3$ UqO with 50 μg of lipopolysaccharide, at intervals of 7 days. Single-cell suspensions were prepared from the spleens of the immunized mice, and the cells were fused with P3U1 myeloma cells, using the conventional polyethylene glycol (PEG) method[68]. Screening of antibodies was performed by three methods, liposome enzyme-linked immunosorbent assay (L-ELISA), denatured ELISA (D-ELISA), and size0exclusion chromatography (SEC)[69]. For L-ELISA, the purified $bo_3$ UqO was reconstituted into liposomes containing biotinyl-PE, and was immobilized on Immobilizer Streptavidin plates (Nunc). High-affinity antibodies that formed stable complexes with the purified $bo_3$ UqO were selected by SEC on a Superdex 200 5/150 column (GE Healthcare). Antibodies that recognized the native conformation of E. coli $bo_3$ UqO were assayed by D-ELISA with SDS-denatured E. coli $bo_3$ UqO. Three selected clones were isolated by the limiting dilution-culture method, and monoclonal hybridoma cell lines producing anti-$bo_3$ UqO antibodies were established. The Fab fragment of an anti-$bo_3$ UqO monoclonal antibody was prepared as described[69]. The purified $bo_3$ UqO proteins were mixed with the Fab fragment for 24 h at 4 °C, and the $bo_3$ UqO–Fab complexes were purified by SEC, followed by incubation with N4. We tested three clones by cryo-EM and selected the one for structural determination.

## Enzyme assay of $bo_3$ UqO

Purified UqO and compounds were incubated on 96-well plates at 25 °C for 60 min in assay buffer (pH 7.4, 50 mM potassium phosphate, 1 mM ethylenediamine-N′,N′,N′,N′-tetraacetic acid (EDTA), 0.1% bovine serum albumin, 0.15% C12M). To start the enzymatic reaction, a reduced 40 μM of UQ-1 was added to the mixture, then the plates were read at 278 nm using a plate reader. The slope was calculated and the enzymatic activity change was exported by comparing it to the DMSO-treated sample as a negative control.

## Cryo-EM data acquisition

A 3 μL sample was applied to a glow-discharged Quantifoil grid (R1.2/R1.3 300 mesh, copper), blotted in 100% humidity at 8 °C, and plunged into liquid ethane using Vitrobot MkIV. The Cryo-EM analysis was initially performed by using a Talos Arctica microscope at 200 kV with a Falcon3 detector at KEK. Data collection was performed by using a Glacios microscope with a Gatan K2 Summit detector in the counting mode at RIKEN SPring-8. Movies were acquired at ×45,000 magnification with an accumulated dose of 50.0 electrons per Å$^2$ over 40 frames. The pixel size was 0.889 Å. The data were automatically acquired by the beam-image shift method using the SerialEM software[70].

## Cryo-EM data processing and model building

Cryo-EM data processing was performed using RELION 3.1[71]. Raw movie stacks were motion-corrected using MotionCor2[72] of RELION's own implementation. The CTF parameters were determined using the CTFFIND4 program[73]. Data processing workflows for UqO (12,388 movie stacks from Glacios) and UqO/compound complex (7173 movie stacks from Glacios) are summarized in Fig. S3A–H, respectively. The final resolution was estimated by gold-standard Fourier shell correlation (FSC) between the two independently refined half maps (FSC = 0.143). For model building against the EM map, an initial model was generated by homology modeling with the MOE program suite (MOLSIS Inc.) and docked into the final map using Chimera[74]. The model was then manually refined using the COOT program[54]. Real-space refinement in the Phenix program was performed with the atomic parameters of hemes revised from the default[52]. A summary of the model parameters and the cryo-EM map statistics are provided in Table S2.

## E. coli growth inhibition assay

E. coli was grown overnight in LB medium and diluted to OD600 of 0.0132. 5 μL of cells was added to each well of the 96-well plate and treated by a compound at concentrations of 128 μg/mL or two-fold serial dilutions in 100 μL LB medium. Plates were then incubated at 37 °C for 24 h without shaking and read at 600 nm using a plate reader to quantify the cell growth. C43 (DE3) stain and C43 (DE3) $\Delta cyo$ were provided by Dr. Yoshio Nakatani and used as wild-type strain and $bo_3$ UqO-knockout strain, respectively. C43 (DE3) $\Delta cyd\Delta appx$ was provided by Dr. Robert Gennis and used for $bo_3$ UqO-dependent strain. As N4 at 32 μg/ml demonstrated growth inhibition from the above assay, we conducted a time- and concentration-dependent viability assay at 32, 128 μg/ml of N4. An inoculum was made of around $1 \times 10^5$ CFU/ml and then incubated at 37 °C without shaking. The surviving colonies were counted at fixed times (12, 24, 48, and 72 h).

## Preparation of $bb_3$ qNOR

Neisseria meningitidis $bb_3$ qNOR was provided by Shiro laboratory. Expression and purification were performed as previously reported with modification[34].

## N. gonorrhoeae culture and minimum inhibitory concentration (MIC)

The N. gonorrhoeae reference strain WHO F[75] and a ceftriaxone-resistant FC428[76] were used for antimicrobial susceptibility testing. N. gonorrhoeae was cultured in GW medium under 5% $CO_2$ at 37 °C without shaking[77]. We outsourced the medium preparation to Research Institute for the Functional Peptides Co., Ltd. The micro broth dilution MIC method was used to quantitatively measure the in vitro antibacterial activity of Q275 against the bacterial strains. The lowest concentration of antibiotic that prevented the growth was interpreted as the MIC. Sodium nitrite (20 mM) was added to the medium to mimic a NO-challenging condition (attack from immune cells). For the colony count assay, we prepared an inoculum of around $5 \times 10^5$ CFU/ml and then incubated it at 37 °C for 24 h without shaking. The surviving colonies were counted.

## Preparation of $norB$-deficient N. gonorrhoeae

Neisseria gonorrhoeae mutants were constructed as described previously[78]. Briefly, in order to construct the N. gonorrhoeae norB-deficient mutant, a 4.3-kb DNA fragment from N. gonorrhoeae FA1090 chromosomal DNA containing the norB gene was amplified with the primers norB-1 and norB-2 by PrimeSTAR Max DNA polymerase (Takara Bio) and cloned into the SmaI site of the pMW119 vector (4.2 kb) (Nippon gene) to construct pHT1729 (8.5 kb). The 6.5-kb DNA region of pHT1729 was amplified with the primers norB-3 and norB-4 by PrimeSTAR Max DNA polymerase and was ligated with a kanamycin-resistance gene (kan) to construct pHT1739. A 3-kb DNA fragment containing the norB allele, in which norB structural gene was replaced with kan gene, was amplified with the primers norB-1 and norB-2 from pHT1739, and transformed into N. gonorrhoeae strains and kanamycin-

resistant clones were selected, resulting in *norB*-deficient mutants. Primers used are,

norB-1TCGAGCTCGGTACCCGATGTAGAACTCTTTATCCACTTTCGGCAG

norB-2CTCTAGAGGATCCCCAGGCGGGCAGCCGCCGTTTCCAACGGTTTG

norB-3GGGAAAACCCTGGCGGTTTTAGCCTGAAAATGGAAACCG

norB-4CATAGCTGTTTCCTGTTTGAGAGCTCCTTTTAATAAATC

## Stopped-flow spectroscopy
To prepare the fully reduced enzyme, an enzyme solution (1 μM of mtCcO or 0.5 μM *bo*₃ UqO), mixed with DMSO or 200 μM N62, was alternately incubated under vacuumed pressure and N₂ atmosphere, then reduced by dithionite. A CO solution was prepared by alternately incubating under vacuumed pressure and a 20% CO atmosphere for mtCcO or 100% CO for *bo*₃ UqO. To investigate the binding of CO to the reduced enzyme, an equal volume of the reduced enzyme solution and the CO solution were mixed in a stopped-flow spectrometer at 4 °C.

## Oxygen consumption rate
An XFe96 Extracellular Flux Analyzer (Agilent) was used to measure the oxygen consumption rate (OCR). Mouse C2C12 cells were seeded in an XFe96 culture microplate ($1.0 \times 10^4$ cells per well) a day before measurement. The OCR of each well was measured in unbuffered DMEM containing 2 mM L-glutamine, and 1 mM sodium pyruvate under basal conditions and in response to 1 μM oligomycin A (Oligo A), 2 μM fluorocarbonyl cyanide phenylhydrazone (FCCP), and 0.5 μM rotenone + 0.5 μM antimycin A (Anti/Rote). OCR for ATP production was calculated as basal OCR−Oligo A OCR using Mito Stress Test Report Generator.

## Cytotoxicity
A microplate LDH assay (Cytotoxicity LDH Assay kit, Dojindo, CK12) was used to determine the cytotoxicity of Q275. The mammalian cells used were rat neonatal cardiomyocytes, cultured in DMEM supplemented with 10% fetal bovine serum. Cells were seeded into a 96-well, flat bottom, tissue culture-treated plate and incubated at 37 °C with 5% CO₂. After 24 hours, the medium was aspirated and replaced with fresh medium containing test compounds. After 24 h of incubation, LDH activity in the supernatant was assessed according to the manufacturers' guidance.

## Phylogenic analysis
To acquire the amino acid sequence from the database, the sequence whose crystal structure was elucidated and registered in the PDB was preferentially selected (PDB ID was shown). For other amino acid sequences, representative ones in the taxon were selected from HCO enzyme types A, B, C, and NOR according to the previous report[41]. These indicated as follows: *Bos taurus* (5B1A_1), *Homo sapiens* (5Z62_1), *Saccharomyces cerevisiae* (6T15_11), *Pseudomonas stutzeri* (3MK7_1), *Rhodobacter capsulatus* (6XKX_4), *Thermus thermophilus* (A2: 2YEV_1, *ba*₃: 1EHK_1), *Escherichia coli* (1FFT_1), *Bacillus subtilis* (*bo*₃: 6KOB_1, *aa*₃: P24010), *Mus musculus* (BBA31677), *Aquifex* (A1: O67935, A2: O67937), *Rhodothermus marinus* (CAC08532), *Paracoccus denitrificans* (P08305), *Yersinia pestis* (WP042593520), *Pseudomonas aeruginosa* (MXH36788), *Snodgrassella alvi* (WP_100091491), *Rickettsiales bacterium* (MBB19300), *Lacrimispora indolis* (VDG67555), *Halobacterium* (WP_010902450), *Natronomonas pharaonis* (CAA71525), *Geobacillus stearothermophilus* (3AYG), and *Pseudomonas aeruginosa* (3OOR).

Multiple amino acid sequence alignment was performed using Clustal Omega v1.2.1 after trimming the N-terminus to align the overall length of the alignment[79]. A default parameter was used, and the conservation of regions essential for the proton pathway (proton channel) in HCOs, such as metal ligands and the catalytic tyrosine residues, was visually confirmed. Neighbor-joining (NJ) trees were inferred with the BioNJ algorithm[80] using PhyML v20160115[81] with the following parameters: 1000 bootstraps, 100 seeds, and correction for multiple substitutions. A majority consensus tree was created and visualized in Dendroscope v3.75[82].

## Statistics and reproducibility
There were no statistics applied except for the analysis of SD for Fig. 5f, g. Experiments were repeated at least two times and confirmed its reproducibility.

## Reporting summary
Further information on research design is available in the Nature Portfolio Reporting Summary linked to this article.

## Data availability
The data that support this study are available from the corresponding authors upon reasonable request. The cryo-EM maps have been deposited in the Electron Microscopy Data Bank (EMDB) under accession code EMD-33293 (apo-*bo*₃ UqO) and EMD-33294 (holo-*bo*₃ UqO). The coordinates have been in the Protein Data Bank (PDB) under accession code 7XMA (apo-mtCcO), 7XMB (holo-mtCcO), 7XMC (apo-*bo*₃ UqO), 7XMD (holo-*bo*₃ UqO). Previously published accession codes used in this work are 5B1A was used for initial phase calculation in X-ray diffraction experiment. 3AG2 was used for MD simulation. For Supplementary Fig. 7, 5Z62, 5B1A, 6GIQ, 6KOB, 6WTI, 1QLE, 2YEV, 1EHK, 6XKW, 5DJQ, 3AYG, 3OOR. Source data underlying Figs. 2B–E, 3E, 4B–I, 5B, D, F, G, S1A–C, S4A–D, S5D, H, J, S6A–D are provided as a Source Data file. Source data are provided in this paper.

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

## Acknowledgements

First and foremost, all authors greatly thank Dr. Tomitake Tsukihara and Dr. Shinya Yoshikawa for their insightful advice and discussion on the project. We thank Ms. Akiko Ogai greatly for the help of protein preparation; Ms. Satomi Kobayashi, Ms. Kanami Inagaki, Ms. Keiko Shingu, and Ms. Satomi Gion for laboratory assistance; Ms. Yuko Okada, Ms. Hisae Kawasaki and Ms. Yukako Kurokawa for secretary assistance; Dr. Yuki Nakamura for the help in the data acquisition in SPring-8 BL26; Dr. Hiroshi Aoyama for the help in the crystal preparation for X-ray experiment; Dr. Takeshi Yokoyama, Dr. Tomomi Uchikubo, Dr. Mikako Shirouzu for the guidance of cryo-EM single particle analysis; Dr. Naruhiko Adachi and Dr. Masato Kawasaki for initial screening for cryo-EM data acquisition of *E. coli* bo3 oxidase preparation (under BINDS 1721); Dr. Robert Gennis and Dr. Sangjin Hong for providing *E. coli* strains and *E. coli* bo3 oxidase construct; Dr. Yoshio Nakatani for providing *E. coli* strains; Dr. Ken Daniel Inaoka and Dr. Kiyoshi Kita for providing *E. coli* strains and advice on bo3 protein preparation; Dr. Vivek Sharma for help on preparation for MD simulation; Dr. Hiroshi Sugimoto for advice on bo3 structural analysis; Dr. Naoki Mochizuki, Dr. Kazu Kikuchi and Dr. Hajime Fukui for comments on the manuscript, Dr. James Pearson for English editing; Ms. C. Shintani for her advice on figure preparation. This research was supported by Platform Project for Supporting Drug Discovery and Life Science Research (Basis for Supporting Innovative Drug Discovery and Life Science Research (BINDS)) from Japan Agency for Medical Research and Development (AMED) under Grant Numbers, JP19am0101113, JP21am0101070, JP20am0101071, JP21am0101082, JP20am0101109, JP20am0101083 (support numbers 1096, 1099, 1721, 2161, 2310 and 2357). The synchrotron radiation experiments were performed at SPring-8 with the approval of the Japan Synchrotron Radiation Research Institute (JASRI) (Proposal No. 2016A2529, 2016B2711, 2017A2555, and 2017B2721, 2018A2721, 2021B2523, 2022A2713). This work was supported by funding from Japan Science and Technology Agency-Core Research for Evolutional Science and Technology (CREST) under Grant Number, JPMJCR14M2 to S.T. AMED under Grant Number, JP19im0210617, JP21ek0109499 and JP22fk0108632 to YShintani. JP21fk0108605 to M.O. Grants-in-Aid for Scientific Research from the Japan Society for the Promotion of Science under Grant Numbers, 21H02914 to S.T., 21K15443 to Y.N., 19H05784 to M.K., and 20K03794 to K.K. Grants-in-Aid for Innovative research "Biometal Sciences" under grant number 20H05497 to Y.Shigeta.

## Author contributions

Y.Shintani conceived the project and performed the mtCcO inhibitor screening. Y.Shintani and Y.N. designed the experimental strategy and analyzed the data. Y.N. performed X-ray crystallography, cryo-EM single particle analysis, in silico compound screening and most of biochemical experiments. S.Yanagisawa

performed resonance Raman spectroscopic experiments and analyzed the data with W.M. and M.K. R.M. performed MD simulation and analyzed the data with K.K., R.H., H.Y., Y.N., and Y.Shigeta. H.S. performed cryo-EM data acquisition and helped data analysis. N.M. and T.K. contributed to crystal data collection and helped X-ray structural analysis. K.S.I. provided purified bovine mtCcO and mtCcO crystals for throughout the project. Y.N. performed in silico compound screening with H.Y. under the supervision of T.H. S.O. and T.M. generated the monoclonal antibodies for bo3 oxidase cryo-EM analysis. K.S., H.T., Y.A., and M.O. provided the *N. gonorrhoeae* strains and performed the MIC determination. Y.I. analyzed and provided advice on the *N. gonorrhoeae* experiments. T.I. performed bo3 biochemical experiment. C.N. performed *Neisseria meningitidis* $bb_3$ qNOR biochemical experiment. H.K. performed OCR measurement using XFe96 Fluxanalyser. S.Yamazaki, T.N., T.Q., and Y.T. performed data analysis including phylogenic analysis and figure preparation. C.G., K.M., T.T., and Y.Shiro provided $bb_3$ qNOR enzyme, the expression construct, and the experimental setting for qNOR. Y.Shintani, Y.N. prepared the original manuscript. S.T. helped with the revision of the original manuscript and acquired funding. All authors reviewed and edited the manuscript.

## Competing interests

The authors declare no competing interests.

## Additional information

[1]Department of Molecular Pharmacology, National Cerebral and Cardiovascular Center, Suita, Osaka, Japan. [2]Department of Medical Biochemistry, Osaka University Graduate School of Frontier Biological Science, Suita, Osaka, Japan. [3]Graduate School of Science, University of Hyogo, Hyogo, Japan. [4]Center for Computational Sciences, University of Tsukuba, Tsukuba, Ibaraki, Japan. [5]RIKEN SPring-8 Center, 1-1-1 Kouto, Sayo, Hyogo, Japan. [6]RIKEN Center for Biosystems Dynamics Research, Yokohama, Kanagawa, Japan. [7]Department of Chemistry, Graduate School of Science, Chiba University, Inage, Chiba, Japan. [8]Department of Bacteriology I, National Institute of Infectious Diseases, Tokyo, Japan. [9]Antimicrobial Resistance Research Center, National Institute of Infectious Diseases, Tokyo, Japan. [10]Center for Basic Education Integrated Learning, Kanagawa Institute of Technology, Atsugi, Kanagawa, Japan. [11]Protein Crystal Analysis Division, Japan Synchrotron Radiation Research Institute, SPring-8, Sayo, Hyogo, Japan. [12]Department of Microbiology and Infectious Diseases, Toho University School of Medicine, Tokyo, Japan. [13]Present address: Structural Biology Division, Japan Synchrotron Radiation Research Institute, SPring-8; Sayo, Hyogo, Japan. [14]These authors contributed equally: Sachiko Yanagisawa, Rikuri Morita. ✉e-mail: shintani.yasunori@ncvc.go.jp

