## [Peer Review File · Nature Communications]

Conserved allosterity buried inside mitochondrial heme-copper oxidases can generate novel antibioticsEditorial Note: Parts of this Peer Review File have been redacted as indicated to maintain the confidentiality of unpublished data.

Reviewers' Comments:

Reviewer #1:

Remarks to the Author:

The authors report an heroic and successful effort to obtain selective small-molecule inhibitors against the bovine mitochondrial COX, the cyt bo3 quinol oxidase from *E. coli* and the qNOR from *N. gonorrhoeae*. Compound T113 was shown by X-ray crystallography to bind in a pocket between TM1, TM2 and TM3 of subunit I and the TM from auxiliary subunit COX7C. The effort to determine the mechanism of inhibition by using resonance Raman spectroscopy and computational methods is not convincing. A more direct approach using kinetics of enzyme function is needed for this. The conclusion is that the inhibition is due to an allosteric narrowing of the oxygen delivery channel. A proper characterization of the inhibition would make a solid contribution in itself.

However, the authors are more interested in using the same approach to obtain inhibitors in bacterial homologues in the same allosteric site, using *E. coli* cyt bo3 and *N. gonorrhoeae* qNOR. Inhibitors are again obtained. There is a major problem that the authors need to address before this work is published. In cyt bo3, the so-called allosteric site is identical to the site where the ubiquinol substrate binds. The inhibitor binds to Asp75 and Arg 71, which also hydrogen bond to the quinol substrate. The authors did not cite the paper reporting the cryoEM structure of cyt bo3 that appeared last year. Furthermore, when selecting a mutant of cyt bo3 to test the effect on their inhibitor of cyt bo3, they chose Glu286, a residue essential for proton transfer and function.

The menaquinol binding site of qNOR was identified by the Shiro group from the Xray structure of the enzyme from *Geobacillus stearothermophilus* and is in the same region of the protein where their inhibitor is thought to bind (TMIII, which is the same as TM1 in cyt bo3). Hence, there is reasonable doubt whether the binding site identified really is an allosteric site in the bacterial quinol oxidases.

The authors have demonstrated that their approach works to obtain novel enzyme inhibitors that could be of therapeutic value. This is certainly useful, though the authors do not make a case for what is unique or new in their approach. In any event, the story they are presenting needs to be revised substantially before publication.

Reviewer #2:

Remarks to the Author:

This is an excellent work and article combining a range of experimental and computational approaches to identify cytochrome oxidase, a key energy producing enzyme, as a potential drug target. Authors applied X-ray crystallography, biochemical and spectroscopy experiments, cryo EM and MD simulations to study the binding of ligands to oxidase.

Authors observed T113 compound binding between subunit I and VIIC, what kind of structural perturbation it causes in structures solved? Does it cause dehydration or hydration of the site?

Authors report binding of N4 molecule to quinol oxidase, closer to the position of T113 in mitochondrial oxidase. Authors should compare the binding site with quinone in quinol oxidase (<https://www.pnas.org/doi/pdf/10.1073/pnas.2106750118>). If inhibitor occupies substrate quinone binding site, it would explain the reasoning for inhibition.

Despite excellent experimental work, the simulations performed are somewhat inferior as far as simulation sampling is concerned. The production runs are 100 ns long, which is very short. Authors must extend this sampling to 1 microsecond. And, then re-perform the analysis they have done.

It would be good to test the simulation results further - for instance, take the snapshot of ligand-

bound MD simulation, remove the ligand and perform MD simulation to see if it relaxes to unbound form. Such tests will strengthen the conclusions.

Authors must put some more effort on explaining the mechanistic reasoning behind inhibition. The field of oxidase has a rich history with molecular mechanism of redox coupled proton pumping rather well understood for bacterial enzymes. In mitochondrial version of oxidase, based on high resolution structural data, a H-channel has been suggested to pump protons by Yoshikawa and colleagues, which does not exist in bacterial and yeast oxidases. Authors should discuss this proton pumping model in the light of their structural data that show perturbations in Asp50/51 and Ser382, which have been discussed as part of proton pumping by H channel.

Secondly, and as said above, H channel is non-functional in yeast and bacterial oxidases and its role in these enzymes has been suggested to be of dielectric well (see works by Peter Rich). Furthermore, recently the Ser382 carrying domain, which authors have observed some conformational changes to occur in has been suggested to affect electron transfer (<https://www.frontiersin.org/articles/10.3389/fmolb.2021.711436/full>). Could ligand binding in mitochondrial oxidase affect electron transfer properties of the enzyme? This aspect can also be discussed briefly next to their model of inhibition.

Minor points;

Line66 - it is unclear what authors mean by amino acid sequences are not similar. They are not same, but also share conservancy of all functionally important residues.

Line 626 - why all water molecules were removed? Do authors see hydration/dehydration of the region in simulations?

Were structural lipids also removed in MD simulation setup?

What was the redox state of metal centers in simulations? And, why that redox state was chosen?

What was the protonation state of amino acid residues? Specially, Glu242 of D channel.

Add references - Gromacs, thermostat and barostat.

Reviewer #3:

Remarks to the Author:

This is an elegant study describing the screening and development of allosteric inhibitors specifically designed to bind at a conserved allosteric site of the heme-copper oxidases (HCO) family. HCOs are highly conserved enzymes that play a central role in the respiratory chain across all domains of life. The different HCOs display distinct structural features, including the presence of an additional helix in Eukaryotes, at the proximity of the identified allosteric binding site. With the increase of antibiotic resistance among bacterial species, it is immanent to develop new antibiotics, targeting key enzymes in central cellular pathways.

The authors used an array of structural biology and biophysical techniques to identify new inhibitors in an elegant synergy. First, they identified an allosteric site in the mammalian HCO. Following this site identification, they have screened for specific inhibitors against HCOs from two distinct bacterial species presenting an alarming increase in antibiotic resistance. Once inhibitors were identified, the authors used a set of experimental and computational methods to shed light on the underlying mechanism governing these allosteric inhibitors' mode of action.

I think that this research paper is of excellent quality and significant findings and, as such, should be published in Nature Communications.

Minor comments:

The manuscript can benefit from language editing to improve readability.

These sentences, for example, do not provide any useful information relevant to this paper; "We have previously found that an endogenous protein, hypoxia-inducible domain family member 1A (Higd1a), directly interacts with mtCcO, positively regulating respiratory activity 24,25. To further extend our findings, we performed random compound screening that modulates mtCcO activity. While we found potential activators for mtCcO, currently in progress for further development," (Page 5)

I would like to see some more details about the developed algorithms and means of selection. "Then, our in-house algorithm integrated them, ranked, and chose the first series of 285 compounds" page 6.

Validation reports for the strictures were not attached.

Responses to the reviewers' comments

We thank all reviewers for their insightful comments regarding our manuscript. According to the reviewers' suggestions, we have performed additional experiments and revised the manuscript. We have carefully addressed all issues raised by the reviewers, and we believe that the manuscript is now much improved and more solid. The revised sentences are marked in blue in the revised manuscript.

Reviewer #1 (Remarks to the Author):

*The authors report a heroic and successful effort to obtain selective small-molecule inhibitors against the bovine mitochondrial COX, the *cyt bo3* quinol oxidase from *E. coli* and the *qNOR* from *N. gonorrhoeae*. Compound T113 was shown by X-ray crystallography to bind in a pocket between TM1, TM2 and TM3 of subunit I and the TM from auxiliary subunit COX7C. The effort to determine the mechanism of inhibition by using resonance Raman spectroscopy and computational methods is not convincing. A more direct approach using kinetics of enzyme function is needed for this. The conclusion is that the inhibition is due to an allosteric narrowing of the oxygen delivery channel. A proper characterization of the inhibition would make a solid contribution in itself.*

Thank you for the valuable and constructive comment. According to the reviewer's suggestion, we performed a stopped-flow experiment in which we could directly assess the binding of carbon monoxide (alternative to molecular oxygen) to the binuclear center (Salomonsson et al. PNAS 204;101:11617-). As shown in the revised Figure 5F (and shown below left), the compound inhibited the CO binding compared to the control (five times slower in Tau). Therefore, we concluded that allosteric inhibition in the oxygen channel plays a major role, not 100% though, in the inhibition of mtCcO. Furthermore, relating to the next question, we also performed a stopped-flow experiment with *bo3* UqO, demonstrating that the inhibitor also slowed the CO binding in *bo3* UqO (Figure 5G,

below right). Thus, we believe that these results strengthen and confirm our characterization of the inhibitory mechanism. We added Figures 5F and G in the revised manuscript and a paragraph in the result section (page 11, line 250 -255, and 261-267).

However, the authors are more interested in using the same approach to obtain inhibitors in bacterial homologues in the same allosteric site, using E. coli cyt bo3 and N. gonorrhoeae qNOR. Inhibitors are again obtained. There is a major problem that the authors need to address before this work is published. In cyt bo3, the so-called allosteric site is identical to the site where the ubiquinol substrate binds. The inhibitor binds to Asp75 and Arg 71, which also hydrogen bond to the quinol substrate. The authors did not cite the paper reporting the cryoEM structure of cyt bo3 that appeared last year.

We thank the reviewer for pointing out this important work. We agree with this point. As shown in PNAS paper reported by Li et al., the quinone-*bo3* UqO structure confirms that N4 occupies the space where the substrate binds. Asp75 and Arg71 are the same amino acids N4 used for molecular interaction as quinone does, suggesting that N4 inhibits *bo3* UqO by obstructing the substrate binding.

In the meantime, we obtained direct evidence that *bo3* UqO inhibitor allosterically narrows the oxygen channel by a stopped-flow experiment as we did on mtCcO (shown in previous page), in addition to the mutant analysis in the oxygen channel (Figure S6C, D).

These findings suggest that the space surrounded by TM0 and TM1- 3 of *bo3* UqO works as a substrate binding site and an allosteric inhibition site we proposed. TM0 is only present in quinol oxidases, including *bo3* UqO. It effectively stabilizes hydrophobic ubiquinol in the transmembrane region so that *bo3* UqO can use as a substrate; the existence of TM0 makes the ubiquinol oxidase family unique. TM0 is not found in other types of HCOs, in which the allosteric site we proposed is distinct from the substrate binding site, as shown in qNOR that does not have TM0.

We thank the reviewer for allowing us a revision. We added the paragraphs in the discussion regarding the mechanisms of inhibition regarding *bo3* UqO (page 14-15, line 309-323), the revised Fig 5F, G and the relevant references.

Furthermore, when selecting a mutant of cyt bo3 to test the effect on their inhibitor of cyt bo3, they chose Glu286, a residue essential for proton transfer and function.

We agree with the reviewer regarding Glu286. According to the papers from the Gennis group, the Glu286 mutant has very little enzymatic activity (none - 6% of WT, *Biochemistry* 1993, 32, 10923-10928, *Biochemistry* 1996, 35, 13673-13680). Although the backbone construct is the same, transformants cultured in our laboratory gave us Glu286Ala (E286A) *bo3* UqO mutant with 8-9 % of WT *bo3* UqO activity. We confirmed that this activity is not from the contamination of quinol oxidation by using an empty vector transformant and repeated experiments. We assumed that it was caused by a difference in the host strain and purification protocol. In addition to the stopped-flow experiments added in the revision, we demonstrated that the N4 effect was deterred by E286A and F117A (Figure S6C), another residue forming the oxygen path. Therefore, we conclude that N4 has the allosteric inhibition on oxygen access of *bo3* UqO in addition to the inhibition of the substrate binding.

The menaquinol binding site of qNOR was identified by the Shiro group from the Xray structure of the enzyme from Geobacillus stearothermophilus and is in the same region of the protein where their inhibitor is thought to bind (TMIII, which is the same as TM1 in cyt bo3). Hence, there is reasonable doubt whether the binding site identified really is an allosteric site in the bacterial quinol oxidases.

The authors have demonstrated that their approach works to obtain novel enzyme inhibitors that could be of therapeutic value. This is certainly useful, though the authors do not make a case for what is unique or new in their approach. In any event, the story they are presenting needs to be revised substantially before publication.

We thank the reviewer for pointing out this critical point. First, we would like to clarify the misunderstanding of the reviewer. In the Shiro group's paper (Matsumoto Y et al. *NSB* 2012), the quinol binding site is the space between TM3 and TM14, not between TM3 and TM4 that is the allosteric site we identified. Mutation analysis of V289 (TM3) and W334 (TM4) suggest that our qNOR inhibitor binds to the allosteric site, not the substrate binding site. We added the indication in the revised Figure 4A.

[Redacted]

These data, together with the evidence of allosteric inhibition in the oxygen channel of mtCcO and *bo*₃ UqO, strongly support our hypothesis that the allosteric site is conserved among the HCO superfamily, although *bo*₃ UqO is a special case where the pocket also works as a substrate binding site.

We all thank the reviewer for giving us the opportunity that we were able to strengthen our findings and approach to searching for allosteric inhibitors. Now we hope that the reviewer agrees with the novelty of our approach to identifying enzyme-specific allosteric inhibitors based on the conserved allostery among the HCOs family.

For Figure 4H

We had difficulty in media preparation for *Neisseria gonorrhoeae* culture. However, we found that outsourcing the media preparation gave us more stable and faster growth after initial submission although the formula was same. We revised the MIC showing clear Q275 effect in Figure 4H.

Reviewer #2 (Remarks to the Author):

This is an excellent work and article combining a range of experimental and computational approaches to identify cytochrome oxidase, a key energy producing enzyme, as a potential drug target. Authors applied X-ray crystallography, biochemical and spectroscopy experiments, cryo EM and MD simulations to study the binding of ligands to oxidase.

We thank the reviewer for appreciating the importance of our study and giving us the opportunity for revision.

Authors observed T113 compound binding between subunit I and VIIC, what kind of structural perturbation it causes in structures solved? Does it cause dehydration or hydration of the site?

Although we carefully analyzed the structure of the T113-CcO complex, there was no significant structural change in the CcO protein found around the inhibitor binding region (subunit I and VIIC) and the oxygen channel in the obtained crystal structure. Water molecules in the inhibitor binding region found in the apo structure was substituted by the inhibitor (dehydration) in the complex structure. In other published mtCcO structures, the water molecules are not in the same position (3ABM and 5B1A shown below), though. Furthermore, we performed apo MD and re-apo MD without the water molecules in the pocket; there was no difference in TM2 or surrounding environment. Therefore, we do not think that the water molecules are critical for structural stabilization.

Authors report binding of N4 molecule to quinol oxidase, closer to the position of T113 in mitochondrial oxidase. Authors should compare the binding site with quinone in quinol oxidase (<https://www.pnas.org/doi/pdf/10.1073/pnas.2106750118>). If inhibitor occupies substrate quinone binding site, it would explain the reasoning for inhibition.

We thank the reviewer for pointing out this important work. We agree with this point. As shown in PNAS paper reported by Li et al., the quinone-*bo*₃ UqO structure confirms that N4 occupies the space where the substrate binds. Asp75 and Arg71 are the same amino acids N4 used for molecular interaction as quinone does, suggesting that N4 inhibits *bo*₃ UqO by obstructing the substrate binding.

In the meantime, we obtained direct evidence that *bo*₃ UqO inhibitor allosterically narrows the oxygen channel by a stopped-flow experiment as we did on mtCcO (Figures shown below, and please see our response to the reviewer 1's 1st comment), in addition to the mutant analysis in the oxygen channel (Figure S6C, D).

These findings suggest that the space surrounded by TM0 and TM1-3 of *bo*₃ UqO works as a substrate binding site and an allosteric inhibition site we proposed. TM0 is only present in quinol oxidases, including *bo*₃ UqO. It effectively stabilizes hydrophobic ubiquinol in the transmembrane region so that *bo*₃ UqO can use as a substrate; the existence of TM0 makes the ubiquinol oxidase family unique. TM0 is not found in other types of HCOs, in which the allosteric site we proposed is distinct from the substrate binding site, as shown in qNOR that does not have TM0.

We thank the reviewer for allowing us a revision. We added the paragraphs in the discussion regarding the mechanisms of inhibition regarding *bo*₃ UqO (page 14-15, line 309-323), the revised Fig 5F, G and the relevant references.

Despite excellent experimental work, the simulations performed are somewhat inferior as far as simulation sampling is concerned. The production runs are 100 ns long, which is very short. Authors must extend this sampling to 1 microsecond. And, then re-perform the analysis they have done.

It would be good to test the simulation results further - for instance, take the snapshot of ligand-bound MD simulation, remove the ligand and perform MD simulation to see if it relaxes to unbound form. Such tests will strengthen the conclusions.

We appreciate the valuable and constructive comment. First, we performed additional simulations with ligand removal, as the reviewer suggested. Ligand removal after ligand-bound MD (Re-apo MD) demonstrated that TM2 relaxed to the initial apo structure, while TM 3 and 4 were stable. However, the change in the oxygen channel did not relax during the re-apo MD. We assume that the effect on TM2 of the inhibitor that is next to TM2 is more direct than on the oxygen channel (shown below).

As our machine resource is limited, we could not manage to perform a 1-microsecond simulation. However, RMSD from the crystal structure during the three independent 100-nanosecond MD for Apo, Holo, and Re-Apo MD was stable, and no sudden change during the 100-nanosecond observation period, suggesting that the effect on TM2 has been stabilized (shown in next page). Thus, we thought that 100-nanosecond MD simulation was sufficient to reveal the structural change caused by the inhibitor in our case.

We thank the reviewer for allowing us to strengthen our simulation. We added the revised Figure 5B, D and paragraphs in the result section (page 10, line 213-217).

Authors must put some more effort on explaining the mechanistic reasoning behind inhibition. The field of oxidase has a rich history with molecular mechanism of redox coupled proton pumping rather well understood for bacterial enzymes. In mitochondrial version of oxidase, based on high resolution structural data, a H-channel has been suggested to pump protons by Yoshikawa and colleagues, which does not exist in bacterial and yeast oxidases. Authors should discuss this proton pumping model in the light of their structural data that show perturbations in Asp50/51 and Ser382, which have been discussed as part of proton pumping by H channel.

Secondly, and as said above, H channel is non-functional in yeast and bacterial oxidases and its role in these enzymes has been suggested to be of dielectric well (see works by Peter Rich). Furthermore, recently the Ser382 carrying domain, which authors have observed some conformational changes to occur in has been suggested to affect electron transfer (<https://www.frontiersin.org/articles/10.3389/fmolb.2021.711436/full>). Could ligand binding in mitochondrial oxidase affect electron transfer properties of the enzyme? This aspect can also be discussed briefly next to their model of inhibition.

We appreciate the valuable comment. We added new paragraphs discussing inhibitory mechanisms as shown below in green (page 16, line 324-342, in the revised manuscript)

and added the relevant references. At least, we can conclude that the allosteric site is conserved among all HCO superfamilies; and it enables us to strategically identify specific inhibitors that could be of therapeutic potential.

Regarding mtCcO, we demonstrated that the inhibitory mechanism on oxygen entry plays a significant role in T113 and N62 inhibition for mtCcO; however other inhibitory mechanisms might also be involved, as in the case we discussed for N4 on *bo*₃ UqO. We found the structural change in Asp50/51 and Ser382 in the mtCcO-T113 crystal structure. We reasoned that it was caused by photoreduction during sample preparation and radiation-induced reduction for the following reasons.

- Change in Asp50/51 and Ser382 is found in the reduced form of CcO structure; however, the simple addition of T113 did not cause the reduction of CcO (Figure S6A, B).
- Laser irradiation during Raman data acquisition caused CcO reduction, suggesting that mtCcO inhibitor lowered the threshold of photoreduction.
- MD simulation with the inhibitor did not cause the structural change in Asp50/51 (Figure S6E, shown above).

These observations did not eliminate the possibility that the binding of T113 induces the structural change found in Asp50/51 and Ser382. The change in these residues might affect proton pumping or electron transfer; Asp50/51 and Ser382 are essential residues for proton pumping, especially in mtCcO, forming H channel as suggested by Yoshikawa et al. (Chem. Rev. 115, 1936–1989 (2015)), although H channel is only found in mtCcO. Rich and colleagues indicated that H channel works as dielectric well (Biochem. Soc. Trans. 45, 813–829 (2017)). Furthermore, the Sharma group recently reported that conformational change in the Ser382 carrying domain affects electron transfer (Front Mol Biosci. 8:711436 (2021)). Thus, perturbation in the region may cause inhibition of proton pumping or electron transfer. Further study is warranted.

Minor points;

Line66 - it is unclear what authors mean by amino acid sequences are not similar. They are not same, but also share conservancy of all functionally important residues.

We apologized for the unclear description. We revised the sentence as the reviewer kindly suggested.

and functionally important residues and thus their core structure is conserved, although the remaining residues are not the same.

Line 626 - why all water molecules were removed? Do authors see hydration/dehydration of the region in simulations?

We usually perform MD in all water molecules removed unless we find a structural disorder. At least no disorder was found in the MD simulation with our setting. Regarding dehydration, please see the response to the 1st point.

Were structural lipids also removed in MD simulation setup?

We used the DOPC bilayer which is available in the Amber lipid14 force field. We usually perform MD without structural lipids because it is challenging for us to calculate force fields for each lipid molecule. No disorder was found in the structure in the current MD simulation with our setting.

What was the redox state of metal centers in simulations? And, why that redox state was chosen?

We used a reduced CcO structure (PDB: 3AG2) with an oxidized force field, although in experiments, we used oxidized CcO and performed enzyme assay in normoxic conditions. We are concerned that the published oxidized structure of CcO has some ambiguity around the binuclear center, which is near the area of interest in our MD. Thus, we decided to use the reduced CcO structure with an oxidized force field, as the reduced CcO structure is more solid.

What was the protonation state of amino acid residues? Specuially, Glu242 of D channel.

In the force field we used, Glu242 was deprotonated. We believe that it represents a form in the catalytic cycle, giving us an insight into the inhibition mechanism of our inhibitor, which is confirmed by the following experiments.

Add references - Gromacs, thermostat and barostat.

Thank you for pointing this out. We added the relevant references in method section.

For Figure 4H

We had difficulty in media preparation for *Neisseria gonorrhoeae* culture. However, we found that outsourcing the media preparation gave us more stable and faster growth after initial submission although the formula was same. We revised the MIC showing clear Q275 effect in Figure 4H.

Reviewer #3 (Remarks to the Author):

This is an elegant study describing the screening and development of allosteric inhibitors specifically designed to bind at a conserved allosteric site of the heme-copper oxidases (HCO) family. HCOs are highly conserved enzymes that play a central role in the respiratory chain across all domains of life. The different HCOs display distinct structural features, including the presence of an additional helix in Eukaryotes, at the proximity of the identified allosteric binding site. With the increase of antibiotic resistance among bacterial species, it is immanent to develop new antibiotics, targeting key enzymes in central cellular pathways.

The authors used an array of structural biology and biophysical techniques to identify new inhibitors in an elegant synergy. First, they identified an allosteric site in the mammalian HCO. Following this site identification, they have screened for specific inhibitors against HCOs from two distinct bacterial species presenting an alarming increase in antibiotic resistance. Once inhibitors were identified, the authors used a set of experimental and computational methods to shed light on the underlying mechanism governing these allosteric inhibitors' mode of action.

I think that this research paper is of excellent quality and significant findings and, as such, should be published in Nature Communications.

We thank the reviewer for appreciating the importance of our study and giving us the opportunity for revision.

Minor comments:

The manuscript can benefit from language editing to improve readability.

These sentences, for example, do not provide any useful information relevant to this paper; “We have previously found that an endogenous protein, hypoxia-inducible domain family member 1A (Higd1a), directly interacts with mtCcO, positively regulating respiratory activity 24,25. To further extend our findings, we performed random compound screening that modulates mtCcO activity. While we found potential activators for mtCcO, currently in progress for further development,” (Page 5)

Thank you for the comment. We agreed the reviewer's point. We modified the sentences indicated as shown in below.

We have previously found that an endogenous protein directly interacts with mtCcO and allosterically modulates mtCcO activity 24,25. This finding led us to perform random compound screening that modulates mtCcO activity; we identified mtCcO inhibitors, chemically distinct from the known inhibitors, including carbon monoxide, nitric oxide (NO), or cyanides.

Also, we asked for language editing from Prof James Pearson, native English speaker, on our revised manuscript for better readability.

I would like to see some more details about the developed algorithms and means of selection. “Then, our in-house algorithm integrated them, ranked, and chose the first series of 285 compounds” page 6.

Thank you for the comment. We added one more reference (J. Chem. Inf. Model. 2012, 52, 1015–1026). In-house algorithm has been previously developed by our co-authors.

Briefly, the computational methods used for virtual screening can be roughly categorized into ligand-based and structure-based methods. When the X-ray structure of a target protein is unavailable, but the active compounds for the protein have been published, ligand-based two-dimensional (2D) and three-dimensional (3D) similarity searches are often used and found useful, especially in the hit-to-lead stage. We have developed the descriptor for the 3D similarity profile to combine the similarity scores from 2D- and 3D-based searches. To this end, we calculated standardized similarity scores from each similarity search algorithm using machine learning from learning data sets. We can obtain better and more diverse compounds from initial hit compounds by integrating the similarity scores from multiple ligand-based search methods.

Validation reports for the structures were not attached.

Thank you for the comment. We added the validation reports for all the structures reported here. We apologize for the inconvenience that we could not attach them at first submission.

For Figure 4H

We had difficulty in media preparation for *Neisseria gonorrhoeae* culture. However, we found that outsourcing the media preparation gave us more stable and faster growth after initial submission although the formula was same. We revised the MIC showing clear Q275 effect in Figure 4H.

Reviewers' Comments:

Reviewer #1:

Remarks to the Author:

The revised manuscript is considerably stronger and has addressed my concerns.

Reviewer #2:

Remarks to the Author:

Authors have answered all my questions and comments, and I am happy with their responses and recommend publication. However, two more points they must mention in the manuscript before it is accepted for publication.

First, although authors did not simulate longer time scales for given reasons and that is understandable, the overall data presented in the article (experiments and simulations combined) is excellent and deserves to be published. Also the new simulation data of re-apo supports their conclusions further. However, they did not see oxygen channel to relax to apo in re-apo simulations, which may be related to simulation time scales. Therefore, I suggest authors to mention in a line or two that further simulations of longer timescales may be needed to clarify molecular mechanistic details.

Second, authors must mention in one line in computational methods the redox states of metal centers and protonation states of amino acid residues. This is important for reproducibility.

Thank you.

Reviewer #3:

Remarks to the Author:

All concerns have been addressed, I support the publication.

Reviewer #4:

Remarks to the Author:

This is a very interesting and exciting study with impressive biochemistry and biophysics. My only comment is that the microbiology is slightly underdeveloped and should be improved. The growth curve assay in *E. coli* (Fig. 3E) is not very information-rich due to the use of a % growth yield assay (with some information missing from the methods section: How long were those plates incubated?). It would be much better to conduct time- and concentration-dependent killing to distinguish between bactericidal effects and bacteriostatic (though these are random categories, this is still standard repertoire for novel antibiotics).

While this is not required for their conclusions, they could also try to generate N4-resistant mutants (which should map to the allosteric inhibition site) to strengthen their biochemistry. But again, just a suggestions, not necessary for this paper.

The *Neisseria* MIC assay (Fig. 4) is fine, but please include number of replicates. Even for this, time- and concentration dependent killing would be superior, but I appreciate the difficulty of growing *Neisseria* in culture.

Responses to the reviewers' comments

We thank all reviewers for their comments on our revised manuscript. We have performed additional experiments and revised the manuscript according to the reviewers' suggestions. We have carefully addressed all issues raised by the reviewers, and we believe that the manuscript is now much improved and more solid. The revised sentences from the latest revision are marked in green.

Reviewer #2 (Remarks to the Author):

Authors have answered all my questions and comments, and I am happy with their responses and recommend publication. However, two more points they must mention in the manuscript before it is accepted for publication.

First, although authors did not simulate longer time scales for given reasons and that is understandable, the overall data presented in the article (experiments and simulations combined) is excellent and deserves to be published. Also the new simulation data of re-apo supports their conclusions further. However, they did not see oxygen channel to relax to apo in re-apo simulations, which may be related to simulation time scales. Therefore, I suggest authors to mention in a line or two that further simulations of longer timescales may be needed to clarify molecular mechanistic details.

Thank you for the valuable comment. According to the reviewer's suggestion, we added new sentences in the discussion shown below in green (page 17, lines 356-359, in the latest revision).

Our MD simulation of 100 ns gave us an important clue to the allosteric mechanism of the mtCcO inhibitor; however, Re-apo MD did not show that the oxygen channel relaxes to apo. Longer MD may be needed to clarify molecular mechanistic details.

Second, authors must mention in one line in computational methods the redox states of metal centers and protonation states of amino acid residues. This is important for reproducibility.

Thank you for the valuable comment. We added the sentences in the methods as shown in below in green (page 42, line 736-741, in the latest revision).

We used a reduced CcO structure (PDB: 3AG2). The parameters for metal centers and amino acids were obtained from earlier study. Briefly, Cu_A and heme a were oxidized and

heme a_3 and Cu_B were reduced. The protonation states of residues were determined by the electrostatic continuum method. The protonation states of the important residues are as follows: all the propionates of hemes a and a_3 were deprotonated, Arg438 and 439 were de-protonated, Asp364 and Glu242 were de-protonated, His290 and His291 were protonated in the δ -position.

Reviewer #4 (Remarks to the Author):

*This is a very interesting and exciting study with impressive biochemistry and biophysics. My only comment is that the microbiology is slightly underdeveloped and should be improved. The growth curve assay in *E. coli* (Fig. 3E) is not very information-rich due to the use of a % growth yield assay (with some information missing from the methods section: How long were those plates incubated?). It would be much better to conduct time- and concentration-dependent killing to distinguish between bactericidal effects and bacteriostatic (though these are random categories, this is still standard repertoire for novel antibiotics).*

We thank the reviewer for appreciating the importance of our study and giving us the opportunity for revision. We apologize that information about the incubation period was lacking. We added this information (24 hours) to the method. According to the reviewer's suggestion, we performed a time- and concentration-dependent viability assay (CFU/ml) to distinguish between the bactericidal and bacteriostatic effects of the compound. We found that the effect of our inhibitor, N4, on *E. coli* is bacteriostatic. We added this assay result in Fig. 3F and below sentences in the result section (page 8, lines 151-153, in the latest revision).

To distinguish between bactericidal and bacteriostatic effects of the compound, we performed a colony count assay and found that the effect of our inhibitor, N4, on *E. coli* is bacteriostatic (Figure 3F).

While this is not required for their conclusions, they could also try to generate N4-resistant mutants (which should map to the allosteric inhibition site) to strengthen their biochemistry. But again, just a suggestions, not necessary for this paper.

Thank you for the valuable comment. We believe that the binding site of N4 on *E. coli* UqO is certain as we determined the binding site by cryo-EM structural analysis. In the future study of antibiotics development for *Neisseria gonorrhoeae* or *Pseudomonas aeruginosa*, which are both in progress now, we would like to generate compound-resistant mutants to verify the binding site further and estimate how fast the bacteria can acquire resistance against the NOR inhibitor.

The Neisseria MIC assay (Fig. 4) is fine, but please include number of replicates. Even for this, time-and concentration dependent killing would be superior, but I appreciate the difficulty of growing Neisseria in culture.

We thank the reviewer for pointing out this point. We apologize that information about the number of replicates was lacking. *Neisseria* growth inhibition assay, we repeated the experiment twice and got the same results; thus, the number of biological replicates is two. We added this information in the Figure legend. Furthermore, we established the knockout of qNOR (*norB*-deficient) in the WT reference strain (WHO F) and ceftriaxone-resistant FC428 strain. In NO-challenging condition (20 mM NaNO₂), *norB*-deficient strains in both WT and FC428 did not grow, further confirming that growth inhibition is mediated via qNOR. To distinguish between bactericidal and bacteriostatic effects, we added colony count after exposure to NaNO₂ and found that targeting qNOR in *N. gonorrhoeae* is bacteriostatic. We only assessed a single time point (24 hours) and *norB*-deficient strains, not Q275 administration. However, given the difficulty of growing *Neisseria* in culture, we believe the results we got from these experiments are informative enough to convince the reviewer.

We added Fig 4I and the comments in the results and the discussion as shown below in green (page 9, lines 183-189, in the latest revision).

We established *norB*-deficient, encoding qNOR, strains in both WHO F and FC428 backgrounds. In NO-challenging condition (20 mM NaNO₂), *norB*-deficient strains in both WT and FC428 did not grow, further supporting that growth inhibition is mediated via qNOR. To distinguish between bactericidal and bacteriostatic effects using these *norB*-deficient strains, we performed colony count after exposure to NaNO₂ and found that targeting qNOR in *N. gonorrhoeae* is bacteriostatic (Fig. 4I).

(page 15, line 307-310, in the latest revision)

Our results suggest that the effects of UqO inhibitor on *E. coli* and qNOR inhibitor on *N. gonorrhoeae* were both bacteriostatic. Although bactericidal action sounds preferable, the superiority of bactericidal action over bacteriostatic has rarely been documented. Further research and development of HCO inhibitors are necessary to reach the clinical arena.

Reviewers' Comments:

Reviewer #2:

Remarks to the Author:

I recommend publication.

Reviewer #4:

Remarks to the Author:

The authors have addressed my suggestions and I have no further concerns (I appreciate the difficulty of working with *N. gonorrhoeae*). Reviewers' Comments:

Reviewer #2:

Remarks to the Author:

I recommend publication.

Reviewer #4:

Remarks to the Author:

The authors have addressed my suggestions and I have no further concerns (I appreciate the difficulty of working with *N. gonorrhoeae*).